Digital preparation and osteology of the skull of Lesothosaurus diagnosticus (Ornithischia: Dinosauria)

Porro Laura B. 1 lporro@rvc.ac.uk
Witmer Lawrence M. 2
Barrett Paul M. 3
1 Department of Comparative Biomedical Sciences, Royal Veterinary College , Hatfield , United Kingdom
2 Department of Biomedical Sciences, Heritage College of Osteopathic Medicine, Ohio University , Athens, OH , United States
3 Department of Earth Sciences, Natural History Museum , London , United Kingdom
Sues Hans-Dieter
Electronic publication date: 2015 Dec 21
Publication date: 2015
Volume: 3
Electronic Location ID: e1494
Received 2015 Oct 28; Accepted 2015 Nov 22
Copyright: ©2015 Porro et al.
Copyright year: 2015
Copyright holder: Porro et al.
License: This is an open access article distributed under the terms of the Creative Commons Attribution License, which permits unrestricted use, distribution, reproduction and adaptation in any medium and for any purpose provided that it is properly attributed. For attribution, the original author(s), title, publication source (PeerJ) and either DOI or URL of the article must be cited.
License URL: https://creativecommons.org/licenses/by/4.0/

Keywords: Anatomy, Computed tomography, Cranium, Dinosaurs, Lower jaw, Visualization

Funding: Ohio University Heritage College of Osteopathic Medicine and the National Science Foundation IBN-0343744 IOB-0517257 IOS-1050154 The Ohio Supercomputing Center LMW acknowledges support from the Ohio University Heritage College of Osteopathic Medicine and the National Science Foundation (IBN-0343744, IOB-0517257, IOS-1050154). The Ohio Supercomputing Center provided additional support. The funders had no role in study design, data collection and analysis, decision to publish, or preparation of the manuscript.

==============================
Several skulls of the ornithischian dinosaur Lesothosaurus diagnosticus (Lower Jurassic, southern Africa) are known, but all are either incomplete, deformed, or incompletely prepared. This has hampered attempts to provide a comprehensive description of skull osteology in this crucial early dinosaurian taxon. Using visualization software, computed tomographic scans of the Lesothosaurus syntypes were digitally segmented to remove matrix, and identify and separate individual cranial and mandibular bones, revealing new anatomical details such as sutural morphology and the presence of several previously undescribed elements. Together with visual inspection of exposed skull bones, these CT data enable a complete description of skull anatomy in this taxon. Comparisons with our new data suggest that two specimens previously identified as Lesothosaurus sp. (MNHN LES 17 and MNHN LES 18) probably represent additional individuals of Lesothosaurus diagnosticus.

Introduction

Ornithischian dinosaurs underwent major taxonomic and ecological radiations during the Jurassic (Sereno, 1997; Butler, Upchurch & Norman, 2008) resulting in diverse craniodental morphologies and, presumably, disparate feeding strategies (e.g., Weishampel & Norman, 1989; Norman & Weishampel, 1991; Sereno, 1997; Barrett, 2014; Mallon & Anderson, 2014). Understanding the evolution of this trophic diversity requires detailed knowledge of skull anatomy (including the potential for cranial kinesis), cranial myology, jaw mechanism, and diet at the base of Ornithischia.

Triassic ornithischians are exceptionally rare, with only two recognized occurrences—Pisanosaurus mertii from Argentina and Eocursor parvus from South Africa, both of which include isolated lower jaws but lack substantial cranial material (Casamiquela, 1967; Bonaparte, 1976; Butler, Smith & Norman, 2007; Butler, 2010). A third potentially Triassic-aged taxon, an indeterminate heterodontosaurid from Argentina (Báez & Marsicano, 2001), may be Early Jurassic in age (Olsen, Kent & Whiteside, 2011). By contrast, the skull of the Early Jurassic heterodontosaurid Heterodontosaurus tucki from South Africa is well known, and has been described in detail on the basis of two almost complete skulls (Crompton & Charig, 1962; Santa Luca, Crompton & Charig, 1976; Norman et al., 2011; Sereno, 2012) and several incomplete skulls (Norman et al., 2011; Porro et al., 2011; Sereno, 2012), including that of a juvenile specimen (Butler, Porro & Norman, 2008). Cranial and lower jaw material is preserved for at least four additional heterodontosaurid taxa from the Early Jurassic of South Africa (Haughton, 1924; Thulborn, 1974; Hopson, 1975; Porro et al., 2011; Sereno, 2012) and one undescribed specimen from the Early Jurassic of western North America (Attridge, Crompton & Jenkins, 1985; Sereno, 1986). Despite their early occurrence and basal position within ornithischian phylogeny (Butler, Upchurch & Norman, 2008), most Early Jurassic heterodontosaurids exhibit cranial and dental specializations atypical of primitive ornithischians, including: a strongly heterodont dentition; closely-packed, chisel-shaped maxillary and dentary ‘cheek’ teeth (some species lack a distinct constriction between the crown and root or a cingulum); heavy tooth wear; a strongly developed coronoid process of the lower jaw; a strongly depressed jaw joint; and extensive fusion of cranial sutures (e.g., Norman et al., 2011; Sereno, 2012). Thus, the skull of Heterodontosaurus is probably not representative of skull morphology in the earliest ornithischians.

Skull material is also known for various Early Jurassic thyreophoran taxa, including Emausaurus (Haubold, 1990), Scelidosaurus (Owen, 1863; Barrett, 2001) and Scutellosaurus (Colbert, 1981; Rosenbaum & Padian, 2000). However, cranial material of Scutellosaurus is disarticulated and fragmentary, and the skulls of Emausaurus and Scelidosaurus display a number of specialisations (such as the presence of curved tooth rows and unusual occlusal relationships) that are unlikely to have been present in more primitive ornithischians. Disarticulated cranial material of Laquintasaura from the Early Jurassic of Venezuela is also available (Barrett et al., 2014), but none of the currently available cranial material is amenable to functional analyses.

By contrast, the Early Jurassic taxon Lesothosaurus diagnosticus Galton, 1978, known on the basis of multiple specimens collected from the Upper Elliot and Clarens formations of South Africa and Lesotho, can serve as a useful model of the early ornithischian condition (Thulborn, 1970; Sereno, 1991; Knoll, 2002a; Knoll, 2002b). It has been incorporated into numerous studies of ornithischian phylogeny (e.g., Norman, 1984a; Sereno, 1984; Sereno, 1986; Sereno, 1999; Cooper, 1985; Maryańska & Osmolska, 1985; Butler, Smith & Norman, 2007; Butler, Upchurch & Norman, 2008), locomotion (Maidment & Barrett, 2011; Bates et al., 2012; Maidment et al., 2014) and feeding (e.g., Thulborn, 1971; Weishampel, 1984; Galton, 1986; Barrett, 1998; Norman, Witmer & Weishampel, 2004; Knoll, 2008). Lesothosaurus possesses a more generalised skull and tooth morphology than that exhibited by heterodontosaurids or thyreophorans, including: low, triangular (‘leaf-shaped’) teeth with a distinct neck and cingulum; coarse denticles on the mesial and distal tooth margins; sporadically developed high-angled marginal tooth wear and evidence of rapid tooth replacement; an inturned, ‘spout-like’ mandibular symphysis; straight tooth rows; and a jaw joint positioned only slightly below the occlusal plane of the tooth row (Thulborn, 1970; Crompton & Attridge, 1986; Norman & Weishampel, 1991; Sereno, 1991; Norman, Witmer & Weishampel, 2004). Nevertheless, the lack of a complete, undistorted skull for Lesothosaurus has limited attempts to reconstruct the morphology and arrangement of the jaw adductor musculature (Thulborn, 1971; Holliday, 2009) or carry out biomechanical analyses of the skull, which could serve as a baseline for comparisons with more derived ornithischian taxa (e.g., Bell, Snively & Shychoski, 2009).

Two skulls (NHMUK PV RU B17 and NHMUK PV RU B23) from the Lower Jurassic Elliot Formation of Lesotho were described by Thulborn (1970) and referred to Fabrosaurus australis. Galton (1978) later designated both specimens as syntypes of a new taxon, Lesothosaurus diagnosticus. These specimens, and other referred material (e.g., NHMUK PV R8501, NHMUK PV R11004, NHMUK R11956), served as the basis for several anatomical descriptions of the L. diagnosticus skull (Thulborn, 1970; Galton, 1978; Norman, 1984b; Weishampel, 1984; Weishampel & Witmer, 1990; Sereno, 1991; Norman, Witmer & Weishampel, 2004). Two additional partial skulls from the Early Jurassic of Lesotho (MNHN LES 17, MNHN LES 18) described by Knoll (2002a) and Knoll (2002b) share numerous characters with L. diagnosticus; however, various anatomical and proportional differences (as well as the larger size of MNHN LES 18) led to these specimens being excluded from L. diagnosticus and assigned to Lesothosaurus sp. Furthermore, the possibility has been raised that MNHN LES 18 may belong to a larger, sympatric neornithischian Stormbergia dangershoeki (Butler, 2005). The skulls of NHMUK PV RUB 23 (Figs. 1A and 1B), MNHN LES 17 and MNHN LES 18 are distorted and missing the anterior snout. NHMUK PV RUB 17 preserves the remains of at least two individuals in three separate blocks: the fully prepared ‘snout’ block contains the anterior ends of the premaxillae and dentaries, and the predentary (Fig. 1E); the ‘palatal’ block contains the bones of the palate, ventral facial region, posterior lower jaws, and ventral braincase (Figs. 1C and 1D). The ‘snout’ and ‘palatal’ blocks pertain to a single individual, while the partially prepared ‘braincase’ block contains the remains of a second individual that also includes several disarticulated skull and postcranial elements (Fig. 1F). Other isolated skull elements (e.g., a maxilla, jugal, squamosal) are also registered as part of NHMUK PV RU B17 and are identical in morphology to those elements preserved in the three blocks. Other specimens figured previously include NHMUK PV R8501, a nearly complete but badly crushed and disarticulated skull, including lower jaws, and NHMUK PV R11956, an articulated but crushed anterior skull with lower jaws, missing its posterior part (Sereno, 1991). One further undescribed specimen, a juvenile skull (NHMUK PV RU C109), provides only limited anatomical information. Unfortunately, no single specimen of Lesothosaurus preserves a skull that is complete, articulated and undistorted.

Figure 1 Syntype specimens of Lesothosaurus diagnosticus examined CT-scanned in this study.

Left (A) and right (B) lateral views of NHMUK PV RU B23. Left (C) and right (D) lateral views of the ‘palatal’ block of NHMUK PV RU B17. Left lateral view of the ‘snout’ block of NHMUK PV RU B17 (E) and ventral view of the ‘braincase’ block of NHMUK PV RU B17 (F).

Computed tomography (CT) is increasingly applied to dinosaur skulls for various purposes: detailed anatomical description (Lautenschlager et al., 2014); digital preparation of fragile material (e.g., Butler et al., 2010; Butler et al., 2012; Porro et al., 2011); reconstruction of disarticulated skulls (Domínguez Alonso et al., 2004; Sampson & Witmer, 2007); imaging of internal cavities such as the endocranial cavity (e.g., Evans, Ridgely & Witmer, 2009; Zelenitsky et al., 2011; Walsh & Knoll, 2011; Knoll et al., 2012; Knoll et al., 2013), semicircular canals (Sereno et al., 2007; Witmer et al., 2008; Walsh et al., 2009) and intracranial sinuses and nasal airways (Sampson & Witmer, 2007; Witmer & Ridgely, 2008; Miyashita et al., 2011; Bourke et al., 2014); and capturing skull morphology for biomechanical analyses (Rayfield et al., 2001; Bell, Snively & Shychoski, 2009; Lautenschlager et al., 2013; Cuff & Rayfield, 2013; Snively et al., 2013; Button, Rayfield & Barrett, 2014). Nearly all of these studies have been based on complete, minimally damaged specimens. This study uses CT scanning and visualization of the Lesothosaurus syntype skulls: (1) to digitally prepare and provide an osteological description of the Lesothosaurus syntype skulls in combination with information from other referred specimens, supplementing and amending previous descriptions (Thulborn, 1971; Galton, 1978; Norman, 1984b; Weishampel, 1984; Weishampel & Witmer, 1990; Sereno, 1991; Norman, Witmer & Weishampel, 2004); and (2) to compare the syntype skulls with material referred to Lesothosaurus sp. (Knoll, 2002a; Knoll, 2002b). The anatomical description is supplemented by 3D PDFs (Figs. S1–S4); as previously noted by Lautenschlager et al. (2014), such documentation permits easier access and inspection of fossil material.

Methods

The two syntype skulls of Lesothosaurus diagnosticus (NHMUK PV RU B17 and NHMUK PV RU B23) were CT-scanned for this study. NHMUK PV RU B23 was CT-scanned at the Center for Quantitative X-ray Imaging at Pennsylvania State University (University Park, Pennsylvania, USA) using an X-Tek micro-focus subsystem of a Varian/BIR Omni-X HD-600 industrial high-resolution CT system at 180 kV and 600 µA. The resulting reconstructions produced 1,107 axial slices with a resolution of 0.15 mm/pixel and a slice thickness of 0.087 mm. Three blocks (described previously) of NHMUK PV RU B17 were scanned at the Ohio University MicroCT Facility (OUµCT, Athens, Ohio, USA) using a General Electric (GE) eXplore Locus CT scanner at 80 kV and 500 µA with 3,600 views and frame-averaging of 8. Reconstruction of the ‘snout’ block produced 583 axial slices at an isotropic voxel size of 0.046 mm; reconstruction of the ‘palatal’ block produced 828 axial slices at an isotropic voxel size of 0.092 mm; reconstruction of the ‘braincase’ block produced 634 axial slices at an isotropic voxel size of 0.092 mm. Additionally, CT scans of MNHN LES 17 were examined to facilitate comparisons between this specimen and the syntype skulls; however, information from this specimen was not used in the anatomical description. This specimen was scanned at OUµCT using the same GE scanner using the same parameters noted above for NHMUK PV RU B17; reconstruction produced 963 transverse slices at a voxel size of 0.092 mm.

CT scans were processed using the 3D visualization software packages Amira 5.3.3 and Avizo 7.1.1 (FEI Visualization Sciences Group, Mérignac Cedex, France). Within the Amira/Avizo segmentation editor, density thresholding was used to separate higher density bone from lower density matrix. Scans were then processed slice-by-slice (interpolating across no more than five slices at a time) to separate bones from each other at sutures, which were identified as lower density areas between bones. In some cases, minerals precipitated within sutures resulting in boundaries with higher density than surrounding bones. Original specimens were used to confirm the location of sutures and to differentiate sutures from post-mortem damage. Individual bones were isolated and separately labeled within the segmentation editor. Three-dimensional surface models (.surf files) of each element were created that could be manipulated in isolation in 3D space; the following anatomical description is based on these surface models (Fig. 2).

Figure 2 Surface models of Lesothosaurus diagnosticus specimens used in this study.

Left (A) and right (B) lateral views, and dorsal view (C) of segmented skull bones of NHMUK PV RU B23. Left lateral view of segmented bones in the ‘snout’ block of NHMUK PV RU B17 (D). Left (E) and right (F) lateral views, and dorsal (G) and ventral (H) views of the ‘palatal’ block of NHMUK PV RU B17. Individual bones are shown in various colours. Anatomical abbreviations: an, angular; ar, articular; bo, basioccipital; bs, basisphenoid; co, coronoid; crt, ceratohyals; d, dentary; ect, ectopterygoid; emf, external mandibular fenestra; f, frontal; frg, braincase fragments; j, jugal; l, lacrimal; lat, laterosphenoid; mx, maxilla; n, nasal; ot, otoccipital; p, parietal; pa, prearticular; pap, palebral; pd, predentary; pf, prefrontal; pl, palatine; pmx, premaxilla; po, postorbital; pt, pterygoid; q, quadrate; qj, quadratojugal; sa, surangular; so, supraoccipital; sp, splenial; sq, squamosal; v, vomer. All scale bars equal 10 mm.

Some portions of the CT scans could not be segmented, partly because the X-ray attenuation properties of the fossil bone and rock matrix were similar enough that contrast was relatively poor. Although the individual bones of the left lower jaw of NHMUK PV RU B23 were successfully isolated, scan resolution of the right lower jaw was too poor to separate individual elements. Scans did not penetrate the interior of NHMUK PV RU B23; it is likely that a complete braincase and palate are present but they cannot be identified. Teeth could not be discerned in CT scans of NHMUK PV RU B23. Maxillary and dentary teeth were identified and segmented in the ‘palatal’ block of NHMUK PV RU B17 but, due to the presence of a high-density precipitate in their pulp cavities, resolution of tooth shape is poor.

Results

Facial skeleton and skull roof

General comments on overall skull morphology are based primarily on NHMUK PV RU B23 (Figs. 2A and 2C), the most complete and least distorted skull in our sample, supplemented with information from the other available specimens. In lateral view, the cranium of NHMUK PV RU B23 is tallest just behind the orbit; the skull roof (frontals, parietal) is gently rounded in lateral profile and the snout tapers smoothly to the premaxillae; there is no break in slope along the snout anterior to the orbits as occurs in Heterodontosaurus (SAM-PK-K1332; Norman et al., 2011; Sereno, 2012). The orbits are circular in outline and are large relative to skull size, representing approximately 36% of basal skull length (i.e., as measured from the anterior margin of the premaxilla to the posterior margin of the basioccipital). The antorbital fossa is sub-triangular in outline, with its apex pointing dorsally, and is relatively small, with a maximum (ventral) length that is approximately 13% of basal skull length. In dorsal view (Fig. 2C), the supratemporal fenestrae are anteroposteriorly longer than mediolaterally wide and have a sub-ovate to sub-triangular outline, whereas the infratemporal fenestrae are sub-rectangular in lateral view and extend for most of the height of the skull. The shape of the external narial opening can be estimated from the partially complete premaxillae of NHMUK PV RU B17 and NHMUK PV R11956 which indicate that the bony narial openings were likely to have been small and sub-ovate in outline.

The craniomandibular joint is depressed relative to the maxillary alveolar margin. In dorsal view, the cranium is widest across the postorbitals, tapering anteriorly to the premaxillae; anterior to this, the shape of the nasals and medial curvature of the maxillae result in a short, strongly pointed muzzle (Fig. 2C). The postorbital portion of the skull has a box-like profile in dorsal view. In occipital view, the skull is widest across the midshafts of the quadrates.

Figure 3 Premaxilla of Lesothosaurus diagnosticus (NHMUK PV RU B17).

Left lateral (A) view of left premaxilla and right lateral (B) view of right premaxilla. Dorsal (C), ventral (D) and posterior (E) views of articulated left and right side elements. Right dorsal oblique view (F) of articulated elements with the bones transparent to visualize the tooth roots, replacement teeth (shown in orange) and canals linking various premaxillary foramina (red). Anatomical abbreviations: a.mx, articulation surface for maxilla; apf, anterior palatal foramen; apmxf, anterior premaxillary foramen; dia, diastema; pmxf, premaxillary foramen. All scale bars equal 10 mm. Scale bar cannot be provided for oblique view (F).

Premaxilla

The premaxilla of Lesothosaurus is composed of a main body with narial, maxillary, posterior and palatal processes (premaxillary shelf). Both the left and right premaxillae are preserved in the ‘snout’ block of NHMUK PV RU B17 (Fig. 3), although both sides lack the maxillary process. Each premaxilla bears six alveoli in addition to several unerupted replacement teeth (two in the right, one in the left) revealed by CT-scanning (Fig. 3F). There is a short edentulous area anterior to the first premaxillary tooth, which is rugose and probably supported a keratinous rhampthotheca (Weishampel & Witmer, 1990; Sereno, 1991; Knoll, 2008; contraThulborn, 1970). The premaxilla forms the ventral margin of the external naris, and a weak excavation is present on the main body ventral to the narial opening, though this forms a smooth slope rather than a distinct fossa (NHMUK PV RU B17; NHMUK PV R8501; NHMUK PV R11956). The anterior premaxillary foramen (Fig. 3A, apmxf) lies at the anteroventral tip of the premaxilla, immediately dorsal to the first alveolus. A second opening, the premaxillary foramen, is positioned more posteriorly and dorsally, at a point just anteroventral to the external naris (Fig. 3A, pmxf). The anterior premaxillary foramen and premaxillary foramen communicate via a deep groove (Sereno, 1991: Fig. 6D). CT scans of NHMUK PV RU B17 demonstrate that the anterior premaxillary foramen is connected to the anterior palatal foramen(Fig. 3D, apf), which opens on the palate, anterior and medial to the first premaxillary tooth; however, the premaxillary foramen does not communicate with the anterior palatal foramen on either side of NHMUK PV RU B17, instead leading to a short, blind-ending canal (Fig. 3F, pmxf) (contra Sereno, 1991). A second palatal foramen described by Sereno (1991) medial to the second and third premaxillary teeth can be seen on the surface of the segmented NHMUK PV RU B17 ‘snout’ block but its route through the premaxilla cannot be traced. The narial processes (‘pre-narial process’ sensu Thulborn (1970)) of the premaxillae are anteroposteriorly broad at their bases and taper slightly as they extend dorsally and slightly posteriorly (Thulborn, 1970). Their dorsal portions are missing in all specimens. Nevertheless, they clearly separate the external nares anteriorly (Fig. 3E), though it is uncertain whether the internarial bar was complete and, if it was complete, how it might have contacted the anterior processes of the nasals. The maxillary processes (‘post-narial process’ sensuThulborn (1970)) are preserved on both sides of NHMUK PV RU B23 (Figs. 2A and 2B, pmx), NHMUK PV R8501 and NHMUK PV R11956; CT scans demonstrate that in NHMUK PV RU B23 the maxillary process of the premaxilla extensively overlaps the dorsal margin of the maxilla and is dorsally overlapped by the ventrolateral edge of the nasals. The process is anteroposteriorly broad ventrally, but narrows slightly as it extends dorsally prior to angling sharply posterodorsally, giving it a kinked appearance in lateral view. The dorsal-most part of the process tapers to a sharp point that is wedged between the nasals and maxilla. A short posterior process of the premaxilla is preserved on the right side of NHMUK PV RU B17 and bears a dorsomedially-directed facet on its dorsal surface (Figs. 3B and 3E, a.mx); although not preserved in articulation in the scanned specimens, this facet probably fitted against the anteroventral corner of the maxilla. The posterior process lacks alveoli, thereby forming a short diastema between the premaxillary and maxillary tooth rows of Lesothosaurus, a feature absent from most previous skull reconstructions (e.g., Thulborn, 1970; Weishampel & Witmer, 1990; Sereno, 1991; Norman, Witmer & Weishampel, 2004). CT scans of NHMUK PV RU B17 confirm that the anterior portions of the premaxillae meet along a dorsoventrally tall, vertical butt joint (i.e., adjacent bones contact at straight, squared-off edges) (Weishampel & Witmer, 1990). Posteriorly, the premaxilla is an inverted ‘L’-shape in transverse section, due to the presence of a dorsally vaulted palate that forms an angle of ∼120° with the alveolar margin of the bone and meets its counterpart at a butt joint along the midline (Fig. 3E). The thickness of the premaxillary palate decreases posteriorly and its contact with the vomer (if any was present) is not preserved.

Maxilla

The maxilla is triangular in lateral view and encompasses all but the posterodorsal portion of the antorbital fossa (Figs. 2 and 4). Both maxillae are preserved in NHMUK PV RU B17 (‘palatal’ block), NHMUK PV RU B23, NHMUK PV R8501 and NHMUK PV R11956 although all elements are slightly damaged. Most of the damage is found either at the anterior margin (e.g., NHMUK PV RU B17) or dorsally (e.g., NHMUK PV R8501); however, the morphology of the entire maxilla is well-represented by these specimens when taken collectively. The number of maxillary teeth varies from a minimum of 12 (in the incomplete left maxilla of NHMUK PV R11956) to at least 15 (in the incomplete right maxilla of NHMUK PV R8501). CT scans of NHMUK PV RU B17 (which possesses 12 left and 14 right maxillary tooth positions) reveal the presence of four replacement teeth in the right maxilla, although it is likely that more teeth are present and cannot be resolved in the scans. In all specimens, the dentition extends for almost the full length of the alveolar ramus (Fig. 4F).

Figure 4 The maxilla of Lesothosaurus diagnosticus.

Surface renderings from NHMUK PV RU B23 (A) and NHMUK PV RU B17 (B–F). The left maxilla in left lateral view (A, B); the left maxilla in medial view (C); the right maxilla in medial view (D); the right maxilla in dorsal view (E); and the left maxilla in posterolateral oblique view (F). The right palatine (blue semi-transparent) is shown in articulation with the right maxilla in (D) and (E). Bones are rendered transparent to visualize tooth roots in (F). Anatomical abbreviations: ad, deep depression in anteroventral corner of antorbital fossa; a.j, articulation surface for jugal; alr, alveolar ramus; a.n, articulation surface for nasal; aof, antorbital fossa; aofe, antorbital fenestra; a.pmx, articulation surface for premaxilla; ascr, ascending ramus; mxsh, medial maxillary shelf; sal, supralveolar lamina. All scale bars equal 10 mm. Scale bar cannot be provided for oblique view (F).

In lateral view, the rounded anteroventral corner of the maxilla is continuous with the ascending ramus of the maxilla dorsally (Fig. 4A, ascr), which forms the anterior margin of the antorbital fossa, and the alveolar margin posteriorly (Fig. 4A, alr). The dorsal margin of the ascending ramus is overlapped by the lateral margins of both the premaxilla and the nasal along short scarf joints (i.e., adjacent bones overlap along beveled edges) (Fig. 4A; Norman, Witmer & Weishampel, 2004). The anterior margin of the ascending ramus forms an angle of approximately 45° with respect to the long axis of the alveolar ramus. The lateral surface of the alveolar margin is generally flat, weakly concave or bears only an indistinct longitudinal swelling; thus a maxillary buccal emargination is effectively absent in Lesothosaurus diagnosticus (Thulborn, 1970; Galton, 1973; Galton, 1978; Sereno, 1991; Knoll, 2008). This area is pierced by a line of up to six small, rounded neurovascular foramina of subequal diameter.

The maxilla forms most of the boundaries of the antorbital fossa (Figs. 4A and 4B, aof); CT scans of NHMUK PV RU B17 and NHMUK PV RU B23 reveal that the medial wall of the fossa is exceptionally thin in comparison to the alveolar ramus and ascending ramus. A distinct trough is present between the medial wall of the antorbital fossa and the supralveolar lamina (Fig. 4E, sal; Sereno, 1991; Witmer, 1997a; Norman, Witmer & Weishampel, 2004). Within the floor of this trough is a foramen that communicates with the external neurovascular foramina, as is typically found in a diversity of dinosaurs (Witmer, 1997a). Both maxillae of NHMUK PV RU B17 and the left maxilla of NHMUK PV RU B23 feature a deeply depressed area in the anteroventral corner of the antorbital fossa (Figs. 4A and 4B, ad) perhaps indicative of an incipient pneumatic recess. Indeed, a “deep and hemispherical” depression occurs in this area in MNHN LES 17 (Knoll, 2002a: 238). An opening is present in this region in Heterodontosaurus (anterior antorbital fenestra of Norman et al. (2011), accessory antorbital fenestra of Sereno (2012)), Haya (maxillary fenestra: Makovicky et al., 2011), Hypsilophodon (Galton, 1974), and in theropods (promaxillary fenestra: e.g., Sampson & Witmer, 2007). The posterodorsal edge of both maxillae in NHMUK PV RU B17 possess a rounded notch that forms the anterior margin of the small antorbital fenestra, which is bounded by the lacrimal posteriorly (Figs. 4B–4D).

CT scans of NHMUK PV RU B23 demonstrate that the anterior process of the lacrimal fits into a slot in the tip of the ascending process of the maxilla (Sereno, 1991); posteriorly, the posterodorsal edge of the maxilla meets the lacrimal along a simple, rounded butt contact. The posteroventral maxilla features a gently everted surface above the tooth row and underlies the anterior ramus of the jugal, which it contacts via a short, dorsomedially-directed scarf joint (Figs. 4A and 4B, a.j; Thulborn, 1970; Sereno, 1991). The maxilla also possesses a short anterior process (Fig. 4A, a.pmx) that is laterally overlapped by the maxillary process of the premaxilla and overlies the palatal process of the premaxilla (Weishampel & Witmer, 1990; Sereno, 1991; Norman, Witmer & Weishampel, 2004).

The internal surface of the maxilla expands above the tooth row to form a longitudinal medial maxillary shelf that has extensive contact with the palatine (Figs. 4C–4E, mxsh) as seen in NHMUK PV R8501 and in CT scans of NHMUK PV RU B17. Sereno (1991): Fig. 4 illustrated a line of ‘special foramina’ (sensuEdmund, 1957) in a referred specimen (SAM unnumbered) and in his skull reconstructions (Sereno, 1991: Fig. 11). However, CT scans of NHMUK PV RU B17 and NHMUK PV RU B23 do not resolve these features, nor is there any unambiguous indication of their presence in other specimens (e.g., NHMUK PV R8501).

Nasal

Both nasals are preserved in NHMUK PV RU B23 (Figs. 2C, 5A and 5B), NHMUK PV R8501 and NHMUK PV R11956, although the majority are damaged: both nasals of NHMUK PV RU B23 and the right nasal of NHMUK PV R11956 are missing their anterior ends and the right nasal of NHMUK PV R8501 is broken posteriorly. However, almost complete left nasals are present in NHMUK PV R8501 and NHMUK PV R11956. A small fragment of the left nasal is present in the ‘palatal’ block of NHMUK PV RUB 17(Fig. 2E, n). In dorsal view, the nasal is an elongate, subtriangular bone that expands laterally at its mid-section to form a short, laterally extending triangular process. The rest of the element tapers to form subrectangular processes anteriorly and posteriorly (Thulborn, 1970; Sereno, 1991). It is approximately 4.5 times as long as it is wide. In transverse section, each nasal is dorsally arched, so that the midline contact between them lies in a shallow depression that extends along the top of the snout. The nasals meet at a straight, vertical butt joint along the midline. The lateral margin of the nasal overlaps the dorsal edges of the premaxilla, maxilla and lacrimal via scarf joints (Fig. 5B, a.pmx, a.mx, a.l); the contact with the lacrimal is very short (Norman, 1984a; Sereno, 1991). The posterolateral margin of the nasal is deeply embayed to accommodate the prefrontal; CT scans demonstrate that the prefrontal extensively overlaps the nasal (Fig. 5A, a.pf). The posterior tip of the nasal overlaps the frontal (Fig. 5B, a.f), as described by Thulborn (1970) and Weishampel & Witmer (1990). In lateral view, the dorsal margin of the nasal is almost flat. Although all of the specimens are either crushed or incomplete, the anterior margin of the left nasal in NHMUK PV R8501 is smoothly concave and clearly formed the posterodorsal margin of the external naris.

Figure 5 Facial bones and skull roof of Lesothosaurus diagnosticus.

All surface renderings are from NHMUK PV RU B23 with the exception of N, taken from NHMUK PV RU B17. Paired nasals in dorsal (A) and ventral (B) views. The left lacrimal in lateral (C), medial (D) and posterior (E) views. The right prefrontal in lateral (F) and medial (G) views. Paired frontals in dorsal (H) and ventral (I) view. Parietal in right lateral (J), dorsal (K) and posterior (L) views. Left jugals in lateral (M) and medial (N) views. Left postorbital in lateral (O), dorsal (P) and medial (Q) views. Jugal (N) is shown transparent to visualize the internal cavity (indicated in red). Elements are shown at different scales. Anatomical abbreviations: a.f, articulation surface for frontal; a.j, articulation surface for jugal; a.l, articulation surface for lacrimal; a.lat, articulation srface for laterosphenoid; a.mx, articulation surface for maxilla; a.n, articulation surface for nasal; a.ot, articulation surface for paroccipital process, part of the otooccipital; aof, antorbital fossa; a.p, articulation surface for parietal; a.pap, articulation surface for palpebral; a.pf, articulation surface for prefrontal; a.pmx, articulation surface for premaxilla; a.po, articulation surface for postorbital; a.so, articulation surface for supraoccipital; a.sq, articulation surface for squamosal; cer?, possible impression for cerebal hemispheres; lc, lacrimal canal; ot, impression for olfactory tract; rug, rugosity on postorbital; stf, supratemporal fossa; stfe, supratemporal fenestra; vr, ventral ridge. All scale bars equal 10 mm.

Lacrimal

The lacrimal has an inverted ‘L’-shape and it separates the antorbital fossa from the orbit. Both lacrimals are preserved in NHMUK PV RU B23 (Figs. 2A and 2B, l), NHMUK PV R8501 and NHMUK PV R11956; only the left element is present in NHMUK PV RUB 17 (Figs. 2E and 2G, l). The short anterior ramus of the lacrimal is rounded in lateral view and inserts into a slot at the tip of the ascending ramus of the maxilla (Figs. 5C and 5D, a.mx) (Sereno, 1991). The lacrimal shaft bears a dorsal facet that articulates with the ventral surface of the prefrontal via a broad ventrolaterally-inclined scarf joint (Figs. 5C–5E, a.pf). The lateral surface of the shaft is gently convex anteroposteriorly while its posterior surface is dorsoventrally and mediolaterally concave (Fig. 5E). The medial surface bears a large triangular depression bounded anteriorly and posteriorly by distinct ridges (Fig. 5D). The opening for the nasolacrimal canal is visible on the posterior surface of the lacrimal shaft (Fig. 5E, lc) (Norman, Witmer & Weishampel, 2004), but its path through the element cannot be fully traced, although the canal likely traversed the lacrimal’s anterior ramus as in Hypsilophodon, Plateosaurus, and most other dinosaurs (Witmer, 1997a). A thin lamina arises from the anteromedial margin of the lacrimal shaft and the ventromedial margin of the anterior process. This sheet of bone forms the posterodorsal portion of the antorbital fossa medial lamina, and also defines the posterior margin of the antorbital fenestra. The ventral ramus tapers posteroventrally to a slender point, overlapping the dorsomedial aspect of the anterior process of the jugal in a long but narrow scarf joint (Fig. 5C, a.j) (Weishampel & Witmer, 1990; Sereno, 1991).

Prefrontal

Both prefrontals are preserved in NHMUK PV RU B23 (Fig. 2C, pf), NHMUK PV R8501 and NHMUK PV R11956. They form the anterodorsal margin of the orbit. Viewed dorsally, the prefrontal is a teardrop-shaped bone with a rounded anterior margin and tapering, slender posterior process (Norman, Witmer & Weishampel, 2004). In transverse section, the prefrontal is flat with a dorsoventrally thick main body and a thin lateral extension that forms the orbital margin; this extension bears a facet on its anterodorsal surface that articulates with the palpebral (Fig. 5F, a.pap; Thulborn, 1970). The main body of the prefrontal extensively overlaps the posterolateral margin of the nasal and the lateral margin of the frontal (Fig. 5G, a.n, a.f; Thulborn, 1970). A short, tapering ventral process arises from the posterolateral corner of the main body and forms an extensive contact with the dorsal surface of the lacrimal shaft along an oblique scarf joint (Fig. 5G, a.l).

Figure 6 Squamosal, quadrate and palpebral of Lesothosaurus diagnosticus.

All surface renderings are from NHMUK PV RU B23. Right squamosal in lateral (A), dorsal (B) and medial (C) views. Left quadrate in lateral (D), medial (E), anterior (F) and posterior (G) views. The left palpebral in lateral (H), dorsal (I) and medial (J) views. Elements are shown at different scales. Anatomical abbreviations: a.ot, articulation surface for paraoccipital process, part of the otoccipital; a.p, articulation surface for parietal; a.pf, articulation surface for prefrontal; a.po, articulation surface for postorbital; a.pt, articulation surface for pterygoid; a.q, articulation surface for quadrate; a.sq, articulation surface for the squamosal; cot, cotylus for head of quadrate; itf, infratemporal fossa; lc, lateral condyle; mc, medial condyle; pqpr, postquadratic process of the squamosal; ptw, pterygoid wing of quadrate; qh, head of quadrate; stfe, supratemporal fenestra. All scale bars equal 10 mm.

Palpebral

Both palpebrals are preserved in NHMUK PV RU B23 (Figs. 2 and 6H–6J) and NHMUK PV R8501; an isolated right palpebral is present in the ‘braincase’ block of NHMUK PV RU B17. They do not traverse the entire diameter of the orbit, although rugosities on the anterior margin of the postorbital suggest that in life the palpebral was connected to the postorbital by the supraorbital membrane. The palpebral shaft is ovoid in transverse section and is bowed laterally in dorsal view (Fig. 6I). The expanded base features short dorsal and anterior processes that articulate with the prefrontal in NHMUK PV RUB 23 (Fig. 6H); there is a point contact between the palpebral and lacrimal (Fig. 6J), but this is not extensive (Thulborn, 1970; Sereno, 1991; contra Weishampel & Witmer, 1990).

Frontal

Both frontals are preserved in articulation in NHMUK PV RU B23 (Figs. 2 and 5H–5I) and NHMUK PV R8501. They are quadrilateral in dorsal view and the frontal is approximately 3.5 times longer than it is wide (as measured at the midlength of the orbital margin). The frontals are widest posteriorly, taper anteriorly, and bear shallow lateral embayments that form the dorsal margins of the orbits (Thulborn, 1970). In lateral view, the frontals are arched anteroposteriorly, giving the skull roof a rounded profile. In transverse section, the anterior half of the frontal is dorsoventrally thickest in its central part (corresponding to the position of the ventral ridge, see below) and tapers in thickness medially and laterally; posteriorly, the ventral ridge merges into the body of the bone. The frontals are dorsally arched in transverse section. The interfrontal suture is straight and the frontals contact each other via a vertical butt joint (Weishampel & Witmer, 1990). The pointed anterior tips of the frontals insert between and underlap the nasals and prefrontals (Fig. 5H, a.n, a.pf). The sharp lateral edge of the frontal forms the central third of the dorsal orbital margin. The orbital margins are smooth and lack the short striations seen in some other small ornithischians. Posterior to the orbit, the frontals expand posterolaterally to meet the postorbital in a complex, undulating suture (Figs. 5H and 5I, a.po). A shallow, well-defined supratemporal fossa excavates the posterolateral corner of the frontal (Thulborn, 1970) and the frontal makes a small contribution to the anteromedial margin of the supratemporal fenestra (Fig. 5I, stf). The posterior margin of the frontal contacts the parietal along an undulating contact; fine interdigitations can be discerned along this contact in NHMUK PV RU B23, but these could not be segmented (Weishampel & Witmer, 1990). The ventral surface of frontal possesses a low rounded ridge (the crista cranii; Fig. I, vr) that extends parallel to the orbital margin and which continues onto the medial surface of the postorbital. This ridge defines the medial margin of the orbital cavity and the lateral margin of the shallowly concave endocranial cavity, in the region of the olfactory tract and cerebral hemispheres (Fig. 5I, ot, cer?). The trough for the olfactory tract extends along the medial part of the ventral surface of the anterior half of the frontals; posteriorly, the depression between the contralateral cristae cranii widens for the area occupied by the cerebral hemispheres. The anterolateral (capitate) process of the laterosphenoid may have contacted the ventral surface of the skull roof near to the frontal-parietal-postorbital contact (Fig. 5I, a.lat).

Parietal

The parietals are preserved in articulation in NHMUK PV RU B23, NHMUK PV R8501 and NHMUK PV R11004, and form the roof of the braincase and the medial and posterior margins of the supratemporal fenestrae (Figs. 2C and 5J–5L). They are strongly dorsally arched in transverse section (Thulborn, 1970), and there is no sagittal crest (Knoll, 2002a; contra Weishampel & Witmer, 1990; Sereno, 1991; Norman, Witmer & Weishampel, 2004). The midline suture between the parietals is visible in CT scans of NHMUK PV RU B23, but is very faint compared to other sutures, suggesting that the elements may have partially fused. Nevertheless, the straight interparietal suture is visible externally in this specimen and also in NHMUK PV R11004. The gently undulating anterior margin of the parietal contacts the posterior edge of the frontal along an interdigitating suture. The short anterolateral process of the parietal contacts the frontal and the medial process of the postorbital via a rounded butt joint (Figs. 5J and 5K, a.f/a.po; Sereno, 1991). The parietals are weakly constricted between the supratemporal openings (Fig. 5K, stfe). The posterior margins of the parietals expand to form prominent posterolaterally extending processes that diverge from the midline of the skull at angles of approximately 45° in dorsal view: together these processes and the straight posterior margin of the main parietal body form a deep embayment that accommodates the dorsal and lateral margins of the supraoccipital (Figs. 5K and 5L, a.so). In dorsal view, the posterolateral process forms the posteromedial margin of the supratemporal fenestra, and in posterior view (Fig. 5L) it is dorsoventrally expanded and overlaps the dorsal and medial surfaces of the squamosal medial process. The ventral margin of the posterolateral process contacts the dorsal margin of the paroccipital process (Fig. 5L, a.ot). It is likely that the ventrolateral margins of the parietal contacted the laterosphenoid, prootic and otoccipital; however, the nature of these contacts cannot be visualized in NHMUK PV RU B23, NHMUK PV R8501 or NHMUK PV R11004 due to disarticulation, crushing, the presence of matrix or difficulty with segmentation. There is some equivocal evidence for the presence of a small opening between the parietal and paroccipital process in NHMUK PV RU B23 (potentially indicative of the posttemporal foramen), but this cannot be confirmed due to deformation and bone loss in the region in this and all other specimens. There appears to be no evidence for the presence of a foramen between the parietal and supraoccipital (contra Sereno, 1991).

Jugal

The jugal is incomplete in almost all specimens, although a nearly complete left jugal missing the ends of some processes is present in NHMUK PV R8501. Combining information from this specimen with partially preserved jugals from the left sides of NHMUK PV RU B17 and NHMUK PV RU B23 (Figs. 2, 5M and 5N) allows most features to be reconstructed. The jugal consists of three processes that arise from a central main body: anterior (maxillary), dorsal (postorbital) and posterior (quadratojugal). In lateral view, the anterior process is slender, being longer than it is tall, and it forms the ventral margin of the orbit. The process tapers anteriorly and extends close to the antorbital fossa though it does not contribute to the margin of the fenestra. Dorsomedially, the anterior process is overlapped by the ventral ramus of the lacrimal (Fig. 5N, a.l). It contacts the posterior process of the maxilla ventrally along a ventrolaterally-directed scarf joint (Figs. 5M and 5N, a.mx). The lateral surface of the anterior process is rounded and continuous with the horizontal shelf of the maxilla. CT scans of NHMUK PV RU B17 reveal a hollow space within the main body of the jugal and a small foramen on the medial surface connects with this space, though a similar foramen cannot be seen in NHMUK PV R8501. These structures likely represent evidence of blood vessels that passed through the floor of the orbit, as similar vascular features are found in a range of dinosaurs and extant diapsids (Sampson & Witmer, 2007). The anterior process is round in transverse section anteriorly, but becomes dorsoventrally expanded and transversely narrow posteriorly, with thickened dorsal and ventral margins. The dorsal process of the jugal forms the ventral part of the postorbital bar and the posteroventral margin of the orbit. It is robust, elongate, tapers dorsally, and bears a long, triangular facet on its anterolateral surface where it is overlapped by the ventral process of the postorbital (Fig. 5M, a.po; Weishampel & Witmer, 1990; Norman, Witmer & Weishampel, 2004). A rounded ridge on its anteromedial surface is continuous with a ridge on the postorbital. The dorsal process of the jugal approaches but does not contact the squamosal (contra Knoll, 2002a; Knoll, 2002b). A partial posterior process is present in NHMUK PV R8501 and NHMUK PV RU B17, which indicates that the depth of the posterior process was greater than that of the anterior process, and that the posterior process formed the ventral margin of the infratemporal fenestra. The medial surface of the jugal is gently concave with a thickened, ridge-like ventral border (Fig. 5N).

Postorbital

Both postorbitals are preserved in NHMUK PV RU B23 (Figs. 2A–2C and 5O–5Q) and NHMUK PV R8501. It is a triradiate bone in lateral view with anterior, posterior and ventral processes. These processes radiate from a subtriangular main body that is laterally convex. All three processes taper distally. The short, stout anterior process joins the frontal via a ‘W’-shaped, undulating contact in dorsal view (Fig. 5P, a.f). The posterodorsal surface of the anterior process is excavated by the anterior corner of the supratemporal fossa (Fig. 5P, stfe). The posteroventral surface of the anterior process bears a small facet that receives the anterolateral process of the parietal and laterosphenoid (Fig. 5Q, a.p/a.lat). The posterior process underlaps the squamosal in an extensive contact, forming the upper temporal bar; it bears a rounded ridge on its ventrolateral surface that dorsally bounds the infratemporal fossa. The ventral process overlaps the dorsal process of the jugal anteriorly and laterally via a long scarf joint, forming the dorsal part of the postorbital bar and the posterodorsal corner of the orbit. The medial surface of the ventral process possesses a ridge that is continuous with ridges on the medial surface of the jugal and on the ventral surface of the frontal (Fig. 5Q, vr). In NHMUK PV RU B23, a slightly rugose area is present along the orbital margin at the junction of the ventral and anterior processes (visible on both postorbitals), probably representing an attachment site for the supraorbital membrane that also would have attached to the palpebral bone anteriorly (Fig. 5O, rug; Maidment & Porro, 2010).

Quadratojugal

Only a small fragment of the left quadratojugal is preserved in NHMUK PV RU B23 (Fig. 2A, qj), attached to the quadrate, but both elements are present in NHMUK PV R8501. The quadratojugal formed the posteroventral corner of the infratemporal fenestra and part of the lateral wall of the adductor chamber. In lateral view, it has an isosceles triangle-shaped outline, with the apex of this triangle pointing dorsally. The anterior and posterior margins are gently concave and the ventral margin is slightly convex, and the posterior margin is closely appressed to the quadrate. The lateral surface is smooth and there is no indication of a paraquadratic foramen. It cannot be determined if the quadratojugal made contact with the squamosal, but a point contact seems plausible as the quadratojugal extended dorsally for approximately half of the height of the quadrate. The contact between the quadratojugal and jugal is not preserved in any specimen.

Squamosal

Both squamosals are preserved in NHMUK PV RU B23 (Figs. 2A–2C and 6A–6C) and NHMUK PV R8501. It is a complexly-shaped, tetraradiate bone with anterior, medial, prequadratic and postquadratic processes. The anterior process is distally expanded (Sereno, 1991) and dorsally and medially overlaps the posterior process of the postorbital to form the upper temporal bar (Fig. 6A, a.po). The dorsal surface of the anterior process is drawn up into a rounded ridge that laterally bounds the supratemporal fossa (Fig. 6B, stfe). The medial process of the squamosal forms the posterior edge of the supratemporal fenestra and laterally and ventrally overlaps the posterolateral process of the parietal (Thulborn, 1970). A sharp lateral ridge between the body of the squamosal and the prequadratic process forms a well-defined sulcus for the origin of the m. adductor mandibulae externus superficialis, in the posterodorsal corner of the infratemporal fenestra. The long, tapering prequadratic process of the squamosal medially and dorsally overlaps the anterior surface of the quadrate (Fig. 6A, a.q; Sereno, 1991). The much shorter postquadratic process overlies the anterior surface of the paraoccipital process (Figs. 6B and 6C, a.ot). The small cotylus receiving the head of the quadrate is deep, cup-shaped and ventrally-directed (Fig. 6A, cot), and is formed by the junction between the pre- and postquadratic processes (Norman, Witmer & Weishampel, 2004). The main body of the squamosal faces dorsally and dorsolaterally, and is small, subtriangular in outline and has a gently convex external surface.

Quadrate

Only the left quadrate of NHMUK PV RUB 23 is preserved, but both are present in NHMUK PV R8501 (Figs. 2A, 2B and 6D–6G). It is composed of a stout sub-vertically inclined shaft, whose anterior surface supports two thin sheets of bone: the medial pterygoid ramus and anterolateral ramus (Norman, Witmer & Weishampel, 2004). Consequently, the quadrate has a ‘V’-shaped horizontal section at mid-shaft . The ventral part of the anterior margin of the anterolateral ramus contacted the quadratojugal (Fig. 6D), whereas the dorsal portion formed a long overlapping contact with the prequadratic process of the squamosal, with these two elements excluding the quadrate from participation in the infratemporal fenestra. The transversely narrow head of the quadrate (Fig. 6F) articulates with the ventral cotylus on the squamosal. The pterygoid ramus is dorsoventrally tall and has a rounded anterior margin (Fig. 6E, ptw); it laterally overlaps the quadrate wing of the pterygoid. The quadrate shaft of Lesothosaurus is strongly anteriorly arched (Weishampel & Witmer, 1990), resulting in the surface of the jaw joint being directed posteroventrally. In posterior view, the shaft is slightly medially-inclined, shallowly excavated, and mediolaterally narrowest above the jaw joint but transversely expanded to form the joint itself (Fig. 6G; Sereno, 1991). There are distinct lateral and medial condyles, which are separated by a shallow groove (Norman, Witmer & Weishampel, 2004; contra Weishampel & Witmer, 1990). The medial condyle is larger than (and ventrally displaced relative to) the lateral condyle.

Palate

The palate is completely preserved (except for the premaxillae) in the ‘palatal’ block of NHMUK PV RU B17 (Figs. 2G and 7). The right palatal complex is largely articulated; the left palatal complex is displaced dorsally relative to the ventral braincase and maxilla, and the ectopterygoid has separated from the pterygoid. The palate is almost certainly present in NHMUK PV RU B23 but cannot be visualized in CT scans due to the presence of dense matrix that limits X-ray penetration.The pterygoids and a left ectopterygoid are present in NHMUK PV R8501, as well as some more anteriorly positioned elements that are badly crushed and difficult to interpret. All of the elements described in this section are based on NHMUK PV RU B17. The palate is dorsally vaulted in transverse section, and dorsally arched in lateral view (Fig. 7B). The maxillary shelves laterally border the internal nares (choana) and appear to exclude the premaxillae. The palatines and anterior processes of the pterygoids form the posterior margin of the internal nares, while the vomers separate them at the midline. The palatal or suborbital fenestra (‘postpalatine fenestra’ of Thulborn (1970) and Sereno (1991)) is very small and bordered by the maxilla, ectopterygoid, pterygoid and palatine; the subtemporal opening is bordered by the posterior tip of the maxilla, jugal, ectopterygoid, pterygoid, quadrate and, presumably, the quadratojugal.

Figure 7 Surface renderings of the right palate of Lesothosaurus diagnosticus from NHMUK PV RU B17 in dorsal (A), lateral (B) and ventral (C) views.

The anterior tip of the vomer has been displaced to the right. Anatomical abbreviations: ect, ectopterygoid; in, internal naris; pl, palatine; plf, palatal fenestra; pt, pterygoid; sto, subtemporal opening; v, vomer. All scale bars equal 10 mm.

Vomer

CT scans demonstrate that the vomers (Fig. 7, v) are elongate elements fused at the midline (Norman, Witmer & Weishampel, 2004) to form the medial margins of the internal nares. They are transversely thin and dorsoventrally tall at their anterior ends (contra Sereno, 1991), being teardrop-shaped in lateral view. The vomers taper posteriorly to a point that lies between the anterior processes of the pterygoids. The vomers do not appear to contact the palatines. As neither the premaxillae nor the anterior portions of the maxillae are preserved in the ‘palatal block’ of NHMUK PV RU B17, it is unclear which of these elements the vomers contacted anteriorly.

Figure 8 Palatal bones of Lesothosaurus diagnosticus.

All surface renderings are from NHMUK PV RU B17. Right palatine in dorsal (A), ventral (B), lateral (C) and medial (D) views. Right ectopterygoid in lateral (E), anterior (F) and medial (G) views (bone shown transparent to reveal hollow cavity within, shown in red). Right pterygoid in lateral (H), dorsal (I) and posterior (J) views. Elements are shown at different scales. Anatomical abbreviations: a.bs, articulation surface for the basipterygoid process; a.ect, articulation surface for ectopterygoid; a.j?, possible articulation surface for jugal; a.mx, articulation surface for maxilla; a.pl, articulation surface for palatine; apr, anterior process of the pterygoid; a.pt, articulation surface for pterygoid; a.q, articulation surface of the quadrate; a.v?, possible articulation surface for the vomer; hl, horizontal lamina of the palatine; m.pt, attachment site of m. pterygoideus ventralis; pnr, palatine pneumatic recess; ptf, pterygoid flange; ptp, pterygoid process of the palatine; qw, quadrate wing of the pterygoid; vl, vertical lamina of the palatine. All scale bars equal 10 mm.

Palatine

Both palatines are complete in NHMUK PV RU B17 (Figs. 2G and 8A–8D), with the right palatine in articulation with its respective maxilla, pterygoid and ectopterygoid. It roofs the posterior palate and forms the posterior margin of the internal nares, although the anterior margin of the palatine is not deeply embayed (contra Thulborn (1970)). The palatine of Lesothosaurus consists of an extensive horizontal lamina and a short vertical lamina (restricted to the posterior half of the element) that are joined laterally (Fig. 8A, hl, vl); thus, the posterior palatine is ‘L’-shaped in transverse section. The vertical lamina possesses a concave facet on its lateral surface that articulates with the medial shelf of the maxilla (Fig. 8C, a.mx); anteriorly, the thickened lateral edge of the horizontal lamina also contacts the maxilla, resulting in a long and extensive suture between these bones (Thulborn, 1970; Sereno, 1991; Norman, Witmer & Weishampel, 2004). The lateral aspect of the vertical lamina also contacts the internal surface of the jugal on the disarticulated left side of NHMUK PV RU B17. Unfortunately, the jugal is missing from the articulated right side and, as a result, the position of the palatine-jugal contact cannot be firmly established (but see Sereno, 1991). A small portion of the vertical lamina may have contacted the internal surface of the ventral ramus of the lacrimal, but this is uncertain. A short, rounded ridge, formed by the vertical lamina, is prominent along the lateral edge of the dorsal surface of the horizontal lamina before birfurcating anteriorly (Fig. 8A). One branch continues along the lateral margin of the element; the other is anteromedially-directed and crosses the dorsal surface of the palatine. As a result, the dorsal surface of the palatine bears two prominent depressions (Fig. 8A, m.pt, pnr). The posterior depression is larger and has been identified as a muscular fossa for the M. pterygoideus dorsalis (Witmer, 1997a); the anterior depression is smaller and has been identified as potentially a palatine pneumatic recess (Witmer, 1997a). The medial margins (Fig. 8D) of the palatines closely approach each other and the midline elements (vomers, parasphenoid), but contact appears unlikely. The posterior tip of the horizontal lamina makes a short contact with the anterior surface of the ectopterygoid (Fig. 8D, a.ect; Sereno, 1991; contra Weishampel & Witmer, 1990). The ventromedial aspect of the horizontal lamina extensively overlies the anterior process of the pterygoid, tapering posteromedially to a spike-like projection.

Ectopterygoid

The ectopterygoid (Figs. 8E–8G) is a hooked, ‘U’-shaped element that connects the pterygoid and palatine with the maxilla (and possibly jugal). The base of the ectopterygoid is broad and extensively contacts the dorsal surfaces of the main body and flange of the pterygoid (not the quadrate ramus, contra Weishampel & Witmer (1990)), contributing to the base and posterior margin of the pterygoid flange. The anterior surface of the ectopterygoid makes a short contact with the posterior tip of the palatine on the right side of NHMUK PV RUB 17 (Fig. 8F, a.pl). The ectopterygoid tapers anterolaterally to a rounded articular facet but, due to disarticulation of this specimen, this surface is free on both sides of NHMUK PV RUB 17 (and also in NHMUK PV R8501). Most likely, it contacted the medial surface of the maxilla, although contact with the jugal cannot be ruled out. The dorsal edge of the ectopterygoid bears a sharp ridge (Sereno, 1991). CT scans reveal a hollow cavity within both ectopterygoids of NHMUK PV RUB 17. Although this cavity might represent an ectopterygoid pneumatic recess, the cavity is fully within the bone and does not open externally, which is a requirement of pneumatic systems (Witmer, 1997a), suggesting that it is simply an open cancellous structure, as seen in many extant diapsids, especially squamates (Figs. 8E–8G; Witmer, 1997a).

Pterygoid

The pterygoid is the largest bone of the palate and links the braincase and sidewalls of the skull (Figs. 2G, 2H and 8H–8J). In consists of a main body, anterior process, a quadrate ramus, and the pterygoid flange. The anterior process is long and formed of a vertical septum of bone that is tallest anteriorly and tapers posteriorly. This process appears to have contacted the vomer medially (Thulborn, 1970), thus contributing to the medial margin of the internal naris (Figs. 8H–8J). The anterior process of the pterygoid underlies the ventral surface of the horizontal lamina of the palatine (Fig. 8I, a.pl). Posteriorly, a thin, horizontal lamina of bone extends laterally from the anterior process and progressively widens and deepens to form the main body of the pterygoid. The left and right main bodies meet in a short, dorsoventrally deep midline butt joint posterior to the pterygoid flange (Figs. 8H–8J, a.pt), resulting in a long, narrow interpterygoid vacuity through which the parasphenoid is visible. As noted above for the ectopterygoid, CT scans reveal hollow cavities within the body of the pterygoid at the level of the pterygoid flange that remain fully within the confines of the bone, breached only by a minute vascular foramen or two. The relatively open cancellous structure of many of the skull bones in NHMUK PV RUB 17 may reflect its subadult or even juvenile status. The quadrate ramus flares posterodorsally and laterally from the main body of the pterygoid and is transversely thin (as well as laterally arched) in transverse section (Figs. 8I and 8J, qw), with a thickened, inturned ventral edge and an undulating posterior margin. The quadrate ramus is overlapped laterally by the pterygoid ramus of the quadrate (Fig. 8H, a.q), forming the medial margin of the subtemporal fenestra. Medial to the base of the quadrate ramus is a deep, posterodorsally-facing concavity that articulates with the basipterygoid processes (Fig. 8J, a.bs). This concavity is delimited ventromedially by a prominent bony projection. The pterygoid flange of Lesothosaurusis triangular in ventral and lateral views, with the apex directed anteriorly (Figs. 8H and 8I, ptf), and lacks strong excavations on its dorsal surface. A rounded concavity on the posterior margin of the pterygoid flange (ventral to the basal articulation) marks the origin of M. pterygoideus ventralis (Fig. 8J, m.pt).

Braincase

NHMUK PV RUB 23 almost certainly preserves a complete braincase but, with the exception of the supraoccipital and otoccipital (opisthotic and exoccipital), it could not be visualized in the CT scans due to high-density matrix within the braincase. NHMUK PV R8501 contains a potentially complete, but partially disarticulated and slightly distorted braincase, whereas the ‘palatal’ block of NHMUK PV RUB 17 preserves a disarticulated basisphenoid (with parasphenoid), basioccipital and a left laterosphenoid/prootic. Another basisphenoid is preserved in the ‘braincase’ block of NHMUK PV RU B17 and an otoccipital is present in NHMUK PV R11004. Many braincase elements are unfused in all of the aforementioned specimens, suggesting juvenile status.

Prootic and Laterosphenoid

A left prootic with a fragment of laterosphenoid is preserved in the palatal block of NHMUK PV RU B17 (Figs. 9A and 9B). As noted by Sereno (1991) a left prootic is also present in NHMUK PV R8501, though the latter is largely obscured by overlying elements and provides very little anatomical information. In NHMUK PV RU B17, the left prootic is well preserved but is out of position such that its lateral surface is now facing dorsally as preserved. Otherwise, the morphology of the prootic is fairly typical. The exposed lateral surface (Fig. 9A) has a fossa for the adductor musclature dorsally, a long pointed process posterodorsally that would have attached to the anterior surface of the otoccipital’s paroccipital process (Fig. 9A, a.pp), and a ventral portion that would have articulated with the basisphenoid. The anterior margin of the prootic is strongly incised for the trigeminal foramen (Fig. 9A, CNV), which would have been completed anteriorly by the laterosphenoid. The contact for the laterosphenoid dorsal to the trigeminal foramen is well preserved as a stout facet (Fig. 9A, a.ls), as in most dinosaurs. A fragment of bone attached to this facet that wraps around to the medial side of the specimen is likely the remnants of the left laterosphenoid (Fig. 9B, ls). The posterior margin the prootic is marked by the long otosphenoidal crest (Fig. 9A, osp) that separated the adductor domain from the middle ear domain (Witmer, 1997b). The crest sweeps anteroventrally from the posterodorsal region (where it probably would have continued onto the otoccipital) down to the basisphenoid region, probably to the region of the basipterygoid process, which is typical for diapsids (Witmer, 1997b). In fact, the dorsolateral wing of the basisphenoid that laterally covers the basisphenoid recess (see below) is probably continuous with the otosphenoidal crest, which is again typical of other diapsids. The foramen for the facial nerve (cranial nerve VII) is just posterior to the otosphenoidal crest and thus within the middle ear cavity, which is by far the most common situation in archosaurs. The anterior margin of the fenestra ovalis (vestibuli) is also preserved posterior to the otosphenoidal crest, posterodorsal to the facial nerve foramen (Fig. 9A, fo). The CT scan data reveal the medial aspect of the bone (Fig. 9B), which is again very conservative. The internal acoustic meatus (Fig. 9B, iam) is an oval depression posterior to the trigeminal foramen that transmits the facial nerve canal, as well as the canals for the two major branches of the vestibulocochlear nerve. The prootic portion of the vestibular pyramid (the conical medial eminence formed by the prootic and otoccipital that houses the vestibule of the inner ear) is well preserved, opening into the substance of the prootic bone from its posterior surface and creating a large vestibular chamber. The CT scan data shows that the lateral (horizontal) semicircular canal opens into this space. Anterior to the vestibular pyramid and dorsal to the internal acoustic meatus is the prootic portion of the fossa for the floccular lobe of the cerebellum, which would have been completed by the otoccipital (Fig. 9B, fr). The preserved fragment of the laterosphenoid is largely uninformative.

Figure 9 Braincase elements of Lesothosaurus diagnosticus.

Surface renderings from NHMUK PV RU B17 (A, B, G–K) and NHMUK PV RU B23 (C–F). Left prootic in lateral (A) and medial (B) views. Left paroccipital process (part of the otoccipital) in lateral (C), posterior (D) and dorsal (E) views. Supraoccipital in posterior (F) view. Basisphenoid in dorsal (G), ventral (H) and left lateral (I) views. Basioccpital in dorsal (J) and ventral (K) views. Elements are shown at different scales. Anatomical abbreviations: a.bo, articulation surface for the basioccipital; a.ls, articulation surface for the laterosphenoid; a.p, articulation surface for the lateral process of the parietal; a.pp, articulation surface for the paroccipital process (otoccipital); a.so, articulation surface for supraoccipital; a.q, articulation surface for quadrate; a.sq, articulation surface for the squamosal; bor, basioccipital recesses; bpp, basipterygoid process; bpr, basipterygoid recess; bt, basal tubera; cp, cultriform process of the parasphenoid; CNV, opening of the trigeminal nerve; CNVII, opening for facial nerve; dg, dorsal groove; ef, endocranial floor; fm, foramen magnum; fo, fenestra ovalis; fr, floccular recess; iam, internal acoustic meatus; ls, displaced fragment of laterosphenoid; nc, nuchal crest; osp, otosphenoidal crest; pf, pituatary process; vie, vestibule of inner ear. All scale bars equal 10 mm.

Otoccipital

The exoccipitals and opisthotics are indistinguishably fused to form the otoccipitals in NHMUK PV RU B23, NHMUK PV R8501 and NHMUK PV R11004. Only the posterior portions could be resolved in scans of NHMUK PV RU B23 (Figs. 9C–9E); anteriorly, the otoccipitals presumably articulated with the basisphenoid and prootic. In posterior view (Fig. 9D), the otoccipitals form the lateral margins of the foramen magnum (Fig. 9D, fm); they flare laterally to gently rounded, non-pendant paroccipital processes, the distal ends of which are slightly dorsoventrally expanded. The anterior surface of the paroccipital process is convex and contacted the postquadratic process of the squamosal. The posterolateral processes of the parietal rested on the dorsal edge of the paraoccipital process (Figs. 9D and 9E, a.p) as did the ventrolateral margins of the supraoccipital (Figs. 9C and 9E, a.so). In NHMUK PV R8501, a small foramen pierces the anterior surface of the paroccipital process (Sereno, 1991: Fig. 13B, labelled post-temporal foramen) and it is plausible that this continued posteriorly to open on the posterior surface of the process as there are indications of a foramen in this area in NHMUK PV R8501, though this canal is not detectable in CT scans of NHMUK PV RU B23. Ventrally, the margin of the process forms a distinct ridge, the otosphenoidal crest. The ventral process of the otoccipital tapers to form the medially concave margin of the foramen magnum prior to expanding lateromedially at its ventral end to form a footplate that articulated with the dorsal surface of the basioccipital. The otoccipital makes a very small contribution to the dorsolateral corner of the occipital condyle (NHMUK PV R8501). Due to the orientation of the otocciptal in NHMUK PV R8501, and damage in NHMUK PV RU B23, the morphology of the jugular foramen cannot be determined in these specimens. However, the anterior margin of the otoccipital in NHMUK PV R11004, although partially damaged and obscured by matrix, does appear to bear at least one small emargination that might represent the posterior margin of the jugular foramen: the dorsal margin of this embayment is formed by a distinct crest, which extends on to the ventral surface of the paroccipital process for a short distance and probably represents the posterodorsal part of the crista interfenestralis. The region where the foramen ovale might be situated is damaged and covered with matrix, so its preservation in this specimen is equivocal (contraSereno, 1991). Three possible openings for cranial nerve exits are visible on the internal surface of the right otoccipital in NHMUK PV R8501 and is seems plausible that these represent the foramina for cranial nerves X (one opening situated anterodorsally) and XII (two openings, situated ventrally) as proposed by Sereno (1991). However, the external openings of these foramina are obscured by overlying elements and matrix in NHMUK PV R8501. Nevertheless, the external openings of at least two, and possibly three foramina, are visible on the lateral surface of NHMUK PV R11004. Details of the semicircular canals are not determinable in the scans of NHMUK PV RU B23 and none of the openings in NHMUK PV R8501 can be confidently identified as semicircular canal openings (contraSereno, 1991).

Supraoccipital

The supraoccipital (Fig. 9F) is a single median element forming the dorsal margin of the foramen magnum and is present in NHMUK PV RU B23 and NHMUK PV R8501. It is trapezoidal in occipital view, being narrowest dorsally and flaring ventrally. In lateral view, the CT scan of NHMUK PV RU B23 reveals an anteriorly tapering, thin sheet that underlies the parietal; it is possible this may represent an anterior process of the supraoccipital but is more likely a broken fragment of the parietal. The posterior surface bears a rounded median nuchal crest that is most prominent dorsally and merges gradually into the main body of the bone, disappearing at approximately midheight (Fig. 9F, nc). The areas to either side of the crest are gently concave. The ventrolateral margins of the supraoccipital bear small facets for articulation with the dorsal margins of the paraoccipital process(Fig. 9F, a.pp), while the dorsolateral margins of the element are bounded by the posterolateral processes of the parietal in posterior view (Sereno, 1991; Norman, Witmer & Weishampel, 2004). Its anterior contacts cannot be resolved in CT scans. The medial (anterior) surface is deeply concave.

Basisphenoid

The basisphenoid is fused to the parasphenoid anteriorly; it presumably joined the prootic, laterosphenoid and otoccipital dorsally, but these contacts are either not preserved, obscured in external view or cannot be visualized in CT scans. Most of this description is based on scan data from the ‘palatal’ and ‘braincase’ blocks of NHMUK PV RU B17(Figs. 9G–9I), but a parabasisphenoid is also present in NHMUK PV R8501. The basisphenoid is tallest and narrowest anteriorly, widening and shortening posteriorly. There is a deep excavation on the dorsal surface of the basisphenoid (Fig. 9G, pf), forming the anterior part of the floor of the endocranial cavity that communicates with the deep, anteroposteriorly narrow pituitary fossa, which opens posterodorsal to the base of the parasphenoid rostrum (cultriform process). On either side of the pituitary fossa, the bone flares posterodorsally and laterally. The sharp ventral edge of this lamina, almost certainly continuous with the otosphenoidal crest noted above with the prootic, forms the anterior and lateral margins of the deep basipterygoid recesses (Figs. 9H and 9I, bpr) of the middle ear cavity. The basipterygoid processes (Figs. 9G–9H, bpp) are short, anteroposteriorly expanded at their distal ends, have a subtriangular cross-section and are rounded at their tips. They are deflected anteroventrally at an angle of ∼55° (relative to the parasphenoid) and laterally at angles of 50° (‘palatal’ block) to 60° (‘braincase’ block) from the midline. Rounded ridges, continuous with the basipterygoid processes, extend posteriorly along the lateroventral margins of the basisphenoid, helping to define a shallow midline depression that extends for the full length of the element. Posteriorly these ridges diverge laterally to form transversely expanded flanges that underlie the basioccipital and form the ventrolateral margins of the basal tubera; these flanges are separated along the midline by a deep, rounded embayment. The internal structure of the bone in NHMUK PV RU B17, as noted for other cranial bones in this specimen, contains relatively large cancellous spaces, presumably filled with marrow in life, that do not open externally. These spaces make it challenging to trace the structures known to traverse the basisphenoid in other diapsids, such as the cerebral branch of the internal carotid artery and the abducens nerves (CN VI). The cerebral carotid artery canals should open into the pituitary fossa, and indeed paired canals open into the posterolateral aspect of the floor of the pituitary fossa from the basiphenoid recesses. This condition of having the cerebral carotids pass through the middle ear to enter the basiphenoid deep to the otosphenoidal crest within the basisphenoid recess is typical for diapsids (Oelrich, 1956; Witmer, 1997b; Sampson & Witmer, 2007; Porter & Witmer, 2015). The canals for the abducens nerves are more difficult to trace, but there is a candidate pair of canals visible in the CT scan data, although their external apertures are difficult to see.

Parasphenoid

The cultriform process of the parasphenoid in the ‘palatal’ block of NHMUK PV RU B 17 is complete and fused to the anterior aspect of the basisphenoid. It projects anteriorly along the midline between the orbits to the midpoint of the palatines. It is shaped like an inverted triangle in transverse section, tapers to a sharp, pointed tip and bears a deep dorsal groove (Norman, Witmer & Weishampel, 2004) for the cartilaginous interorbital septum. CT scans reveal that this groove continues into the body of the basisphenoid, although it does not connect with the pituitary fossa. The long axis of the cultriform process is situated at the same level as the long axis of the basisphenoid, just dorsal to the bases of the basipterygoid processes, and is not dorsally or ventrally offset.

Basioccipital

A disarticulated basioccipital is preserved in the ‘palatal’ block of NHMUK PV RU B17 (Figs. 9J and 9K); NHMUK PV R8501 also includes a basioccipital that is partially obscured by surrounding cranial elements. In posterior view, the occipital condyle is shaped like a rounded, inverted triangle and formed the ventral margin of the foramen magnum. Anterior and lateral to the condyle are the deep concavities of the basioccipital recesses, which are separated by a low, sharp ridge at the midline. A pair of hook-shaped processes from the lateral aspect of the basioccipital form the dorsolateral margins of the tubera. The dorsal surface of the posterior basioccipital is slightly depressed; anteriorly, a low median ridge and paired lateral ridges form two distinct depressions. Facets on the anterolateral surfaces of the basioccipital mark its contact with the basisphenoid; as the basioccipital and basisphenoid are disarticulated in the ‘palatal’ block and the former is lost in the ‘braincase’ block, it appears these elements were not strongly joined. The basioccipital contacted the otoccipitals dorsolaterally, via two large crescentic facets (NHMUK PV R8501).

Lower jaw

The lower jaw of L. diagnosticus has a nearly straight ventral margin and is only slightly upturned at its anterior end (Fig. 2). The dentary forms over half of the length of the lower jaw in lateral view, and there is a well-developed coronoid eminence, though this is not extended dorsally into a tall, distinct coronoid process. The jaw joint is slightly depressed relative to the alveolar margin (Fig. 2A). There is an anteroposteriorly elongate external mandibular fenestra between the dentary, surangular and angular (Figs. 2A, 2E and 2F).

Figure 10 Predentary and dentary of Lesothosaurus diagnosticus.

Surface renderings are from NHMUK PV RU B17: the predentary and anterior right dentary is found in the ‘snout’ block while the posterior right dentary is preserved in the ‘palatal’ block. Predentary in dorsal (A), lateral (B) and ventral (C) views. Anterior and posterior portions of the right dentary in lateral (D) and medial (E) views. Elements are shown at different scales. Anatomical abbreviations: a.an, articulation surface for angular; a.co, articulation surface for coronoid; a.d, articulation surfaces for dentary; af, anterior foramen; a.pd, articulation surface for predentary; a.sa, articulation surface for surangular; a.sp, articulation surface for splenial; be, buccal emargination; dp, dorsal process of the dentary; emf, external mandibular fenestra; fo, foramina; lp, lateral process of the predentary; vp, ventral process. All scale bars equal 10 mm.

Predentary

A predentary is preserved in the ‘snout’ block of NHMUK PV RU B17 (Figs. 2D and 10A–10C) and in NHMUK PV R8501; both examples are preserved in articulation with the dentaries. As noted in previous descriptions, it is shaped like an arrowhead in ventral view, with a long median ventral keel and slightly shorter lateral processes separated by deep embayments that accommodate the anterior ends of the dentaries (Fig. 10C, a.d). The oral margin of the predentary is smooth and straight in lateral view and the anterior tip is not curved dorsally (Fig. 10B). In transverse section, the anterior predentary is shaped like an inverted triangle with a flat occlusal surface; posteriorly, it becomes ‘V’-shaped. Two prominent foramina are visible in lateral view: the first at the junction between the lateral and ventral processes, and the second within the lateral process (Fig. 10B, fo). The presence of high-density precipitates at the predentary-dentary joint makes tracing these openings into the body of the predentary difficult; however, the abundance of these precipitates suggests that the predentary and its overlying rhamphotheca was richly supplied with blood vessels and nerves. The ventral keel is triangular in transverse section, with its dorsal apex fitting between the anterior ends of the dentaries (Norman, Witmer & Weishampel, 2004). The lateral processes become mediolaterally thin and laterally overlap the dentaries.

Dentary

Partial or complete dentaries are known from many specimens (Fig. 2), including NHMUK PV R8501 (missing only the anterior end of the right dentary), NHMUK PV RU B17 (though divided between the ‘snout’ and ‘palatal’ blocks), NHMUK PV RU B23 (posterior two-thirds of both dentaries, although only the left dentary, and other individual bones of the lower jaw, could be resolved in CT scans), and NHMUK PV R11956 (partial posterior parts of both dentaries). The left dentary of NHMUK PV RU B17 (including portions in the ‘snout’ and ‘palatal’ blocks) preserves 12 tooth positions and the right dentary preserves 17 tooth positions; more teeth are likely to be present but cannot be resolved in scans. The dorsal and ventral margins of the dentary are parallel throughout its length (Figs. 10D and 10E; Sereno, 1991; Weishampel & Witmer, 1990; Norman, Witmer & Weishampel, 2004). The anterior end of the dentary tapers abruptly to a rounded point (Fig. 10D) and twists about its long axis so it meets its opposite ventrally, forming a distinct ‘spout-shaped’ symphysis. The anterior dentary bears a convex, ventromedial facet that contacts the ventral predentary process (Fig. 10E, a.pd) and a smaller, flat dorsolateral facet for the lateral predentary process (Fig. 10D, a.pd). The contact between the lateral predentary processes and the dentaries are tight while the contact between the dentaries and the ventral predentary process are patent. CT scans reveal that the anterior dentaries meet each other at an anteroposteriorly short, flattened midline contact restricted to the lower third of the element (Fig. 10E, a.d). The lateral surface of the anterior dentary bears a prominent foramen (the ‘anterior dentary foramen’ of Sereno (1991)) between the lateral and ventral predentary processes; three additional large foramina are exposed on the ventrolateral surface of the anterior dentary (Fig. 10D, fo). All of these large foramina (as well as a number of smaller openings on the ventromedial aspect of the anterior dentary) can be traced to a precipitate-filled Meckelian canal that becomes taller posteriorly. CT scans confirm a short, edentulous area of the dentary between its contact with the lateral predentary process and the first dentary tooth.

Unlike the flat ventrolateral surface of the maxilla, the external surface of the dentary is convex and the dentary teeth are inset (Fig. 10D, be; Thulborn, 1970; Sereno, 1991). Posteriorly, the dentary bifurcates into long tapering dorsal and ventral processes that overlap the surangular and angular, respectively (Fig. 10E, dp, vp; Thulborn, 1970; Norman, Witmer & Weishampel, 2004). The dorsal process forms the anterior border of the coronoid eminence. A deep embayment between these processes forms the anterior half of the external mandibular fenestra (Fig. 10D, emf). The middle and posterior parts of the dentary are ‘C’-shaped in transverse section, laterally arched and have thickened dorsal and ventral margins. The ventromedial edge of the dentary meets the ventrolateral margin of the splenial in a rounded butt joint along most of its length (Fig. 10E, a.sp). A long anterior process of the angular contacts the ventromedial aspect of dentary. The posterior third of the dorsomedial margin of the dentary contacts the lateral surface of the coronoid(Fig. 10E, a.co). Replacement foramina on the medial surface of the dentary cannot be resolved in CT scans though they are clearly visible in NHMUK PV R8501. In medial view, the dentary forms the anterior boundaries of the internal mandibular fossa.

Figure 11 Lower jaw bones of Lesothosaurus diagnosticus.

Surface renderings are from the ‘palatal’ block of NHMUK PV RU B17 (A, B, E–K) and NHMUK PV RU B23 (C, D). Right splenial in lateral (A) and medial (B) views. Left coronoid in lateral (C) and medial (D) views; anteroventral margin of this element is damaged. Left surangular in lateral (E), medial (F) and dorsal (G) views; ventral margin of the anterior process is damaged. Left angular in lateral (H) and medial (I) views. Left prearticular in lateral (J) and medial (K) views; the anteroventral margin and posterior tip of this element are missing. Left articular in lateral (L), dorsal (M) and medial (N) views. Elements are shown at different scales. Anatomical abbreviations: a.an, articulation surface for angular; a.ar, articulation surface for articular; a.co, articulation surface for coronoid; a.d, articulation surface for dentary; a.pa, articulation surface for prearticular; a.sa, articulation surface for surangular; a.sp, articulation surface for splenial; cmj, craniomandibular joint; dp, dorsal process; dr, dorsal ridge; emf, external mandibular fenestra; imf, internal mandibular fenestra; lr, lateral ridge; m.ae, attachment site of m. adductor externus superficialis group; maf, mandibular adductor fossa; mf, medial flange; rp, retroarticular process; sf, surangular foramen; vp, ventral process. All scale bars equal 10 mm.

Splenial

The splenial is a transversely flattened sheet of bone forming much of the medial aspect of the lower jaw and encloses the Meckelian canal medially; it is preserved on both sides of the ‘palatal’ block of NHMUK PV RU B17 and can be visualized on the left side of NHMUK PV RU B23 (Fig. 2). In CT scans the dorsal and ventral margins are slightly thickened and the ventral margin is inturned to contact the ventral edge of the dentary in a simple butt joint along much of its length (Fig. 11A, a.d). In medial view, the anterior margin of the splenial is gently rounded; it approaches but does not reach the symphysis (Fig. 11B). Posteriorly, the splenial bifurcates into a short, rounded dorsal process and a long, tapering ventral process. A rounded notch between the processes contributes to the anterior half of the small internal mandibular fenestra (Fig. 11B, imf). The dorsal process of the splenial laterally contacts the anterodorsal process of the prearticular; the dorsal margin of this process meets the ventral edge of the coronoid in a simple butt joint (Fig. 11A, a.co). The longer ventral process of the splenial extensively underlaps the anterior process of the angular; posteriorly, it also underlaps the main body of the prearticular (Fig. 11A, a.pa).

Coronoid

The coronoid is preserved on both sides of the ‘palatal’ block of NHMUK PV RU B17 and the left side of NHMUK PV RU B23 (Fig. 2). Anteriorly, it is a mediolaterally flattened strip of bone applied to the dorsal margin of the dentary (Fig. 11C, a.d) with its ventral margin resting on the dorsal edge of the splenial (Fig. 11D, a.sp); it extends along the posterior third of the dentary. It increases in height and width posteriorly, forming the highest point of the coronoid eminence and developing a lateral process that overlies the dorsal ramus of the surangular (Fig. 11C, a.sa). A ventral tab of the coronoid dorsally and medially overlaps the anterodorsal process of the prearticular (Fig. 11C, a.pa). There is a rounded ridge on the dorsal aspect of the coronoid that marks an area of muscle attachment.

Surangular

Both surangulars are preserved in NHMUK PV RU B23, the ‘palatal’ block of NHMUK PV RU B17 (Fig. 2) and NHMUK PV R8501; additionally, an isolated but well-preserved right surangular is present in the ‘braincase’ block of NHMUK PV RUB 17 (Fig. S4). The tapering dorsal ramus of the surangular underlies the dorsal process of the dentary (Figs. 11E and 11G, a.d). Additionally, it features a dorsomedial facet that underlaps the lateral process of the coronoid (Figs. 11E–11G, a.co) and, on the left side of the ‘palatal’ block (NHMUK PV RU B17), the medial margin of the dorsal surangular ramus has a short contact with the posterodorsal process of the prearticular (Fig. 11F, a.pa). Posteriorly, the surangular increases in dorsoventral height; in transverse section, the element is laterally arched, with the dorsal margin strongly inturned and thickened. In external view, this results in a flattened area (Figs. 11E and 11G, m.ae) on the dorsolateral aspect of the surangular that is bounded dorsally and medially by a strong ridge (Figs. 11E and 11F, dr); this ridge is continuous anteriorly with the dorsal ridge of the coronoid. Together, this flat area and ridge mark areas of attachment for portions of the M. adductor mandibulae externus group. The posterior half of the lateral surface of the surangular bears a longitudinal ridge(Figs. 11E and 11G, lr); the surangular foramen opens immediately below the posterior end of this ridge (Fig. 11E, sf). Below the lateral ridge, the ventrolateral aspect of the surangular is overlapped by the angular (Fig. 11E, a.an); the isolated surangular in the ‘braincase’ block clearly exhibits a large facet for this contact. The anteroventral edge of the surangular forms the posterodorsal margin of the external mandibular fenestra. The dorsal margin of the surangular is convex in lateral view (Sereno, 1991) and the bone decreases in height posteriorly. Immediately anterior to the jaw joint, there is a robust, medial extension (‘medial flange’ of Thulborn (1970)) of the surangular with a strongly concave anterior surface (Figs. 11F and 11G, mf). This flange forms the posterior wall of the adductor fossa, contacts the dorsoventral expansion of the prearticular, contacts the anterior tip of the articular, and forms the anterior margin of the jaw joint (Fig. 11F). Posteriorly, the surangular is deflected medially and becomes a transversely thin sheet of bone that laterally overlaps the articular, forming the retroarticular process (Figs. 11E–11G, rp). A faint ridge divides the lateral surface of the retroarticular process into dorso- and ventrolateral surfaces. In medial view, the dorsal margin of the surangular is slightly inturned to form the border of the internal mandibular fenestra.

Angular

Both angulars are preserved in NHMUK PV RU B23, the ‘palatal’ block of NHMUK PV RU B17 (Fig. 2) and NHMUK PV R8501; a fragment of a right angular (attached to the isolated surangular) and an articulated left angular, articular and prearticular are also preserved in the ‘braincase’ block (Fig. S4). A tapering anterior process extends into the Meckelian canal and overlaps the internal surfaces of both the dentary and splenial (Figs. 11H and 11I, a.d, a.sp). Posteriorly, the angular increases in height and becomes ‘L’-shaped in transverse section. The vertical lamina externally overlaps the surangular while the horizontal lamina underlies the prearticular (Fig. 11I, a.pa). The anterodorsal margin of the angular forms the posteroventral margin of the external mandibular fenestra (Fig. 11H, emf). In medial view, the angular forms the ventrolateral border of the internal mandibular fenestra.

Prearticular

Both prearticulars are preserved in NHMUK PV RU B23 (although only the left element could be visualized in CT scans) and in the ‘palatal’ block of NHMUK PV RU B17 (Fig. 2); an additional left prearticular is preserved in the ‘braincase’ block (Fig. S4). Although no single element is completely preserved, collectively the entire prearticular is represented. Posteriorly, the prearticular is a thin sheet of bone that extensively overlaps the medial surfaces of the articular and medial expansion of the surangular (Fig. 11J, a.ar, a.sa). This sheet of bone is strongly depressed laterally and marks a possible muscle attachment site. The prearticular decreases in height anteriorly and lies in a trough formed by the angular and splenial to form the floor of the mandibular adductor fossa (Fig. 11K). Anteriorly, the prearticular becomes transversely thin and dorsoventrally tall to form a robust anterodorsal process that articulates with the ventral tab of the coronoid (Fig. 11K, a.co), medial margin of the surangular (Fig. 11J, a.sa), and dorsal process of the splenial (Fig. 11K, a.sp), forming the anterodorsal margin of the internal mandibular fenestra.

Articular

Both articulars are preserved in NHMUK PV RU B23 and in the ‘palatal’ block of NHMUK PV RU B17 (Fig. 2); an additional left prearticular is preserved in the ‘braincase’ block(Fig. S4). The articular is nearly as wide as it is long; its dorsoventral height is approximately half its mediolateral width. In dorsal view the articular is widest across its centre and tapers anteriorly and posteriorly (Fig. 11M). It is held in a cup formed by the prearticular, angular and surangular (Figs. 11L and 11N). The tapering anterior process extends medial and ventral to the medial extension of the surangular (Fig. 11L); the posterior processes of the articular and surangular form the retroarticular process. The dorsal surface forms the jaw joint (Fig. 11M, cmj); there is no median ridge separating the surfaces for the lateral and medial condyles of the quadrate. A strong ridge traverses the articular and forms the anterior margin of the joint surface.

Discussion

Digital preparation has clarified the cranial anatomy of Lesothosaurus diagnosticus as well as revealing new features, allowing fresh comparisons with two additional ornithischian specimens from the Upper Elliot Formation of Lesotho (MNHN LES 17 and MNHN LES 18). Knoll (2002a) and Knoll (2002b) compared these skulls with those of Lesothosaurus diagnosticus and concluded that, despite sharing many features with the syntypes, MNHN LES 17 could be distinguished from them. Consequently, he assigned this specimen to Lesothosaurus sp. Differences between MHHN LES 17 and the syntypes identified by Knoll (2002a) include:

1. Possession of a deep, hemispherical depression in the anteroventral corner of the antorbital fossa in MNHN LES 17 that was thought to be absent from the syntypes of L. diagnosticus. However, CT scans demonstrate that this depression is present on both sides of NHMUK PV RU B17 (‘palatal’ block) and NHMUK PV RU B23.

2. The antorbital fenestra was described as reniform in outline and relatively larger in MNHN LES 17 than in L. diagnosticus. However, crushing has caused anterior displacement of the lacrimal on both sides of NHMUK PV RU B23, effectively closing this fenestra. Its true size and shape can be better appreciated in segmented scans of the left side of NHMUK PV RU B17 (‘palatal’ block), which shows that this structure was similar in both MNHN LES 17 and L. diagnosticus.

3. The frontal-nasal suture is flush and the prefrontal-nasal sutures are offset in MNHN LES 17, whereas in NHMUK PV RU B23 the prefrontal-nasal contacts are flush while the frontals are ventrally offset. Observations of extant crocodilian and squamate skulls suggest that these sutural contacts on the external skull roof were likely flush in life and that the offsets in both specimens are due to post-mortem deformation (Knoll, 2002a).

4. The ventral branch of the postorbital is straighter in MNHN LES 17 than in NHMUK PV RU B23. However, both skulls have suffered some degree of deformation and both postorbitals are incomplete in MNHN LES 17.

5. The quadratojugal is separated from the quadrate condyle in MNHN LES 17, whereas it closely approaches the level of the jaw joint in L. diagnosticus. Re-examination of this area indicates that this region is heavily damaged in both MNHN LES 17 and most L. diagnosticus specimens, suggesting that this intepretation of this feature may be ambiguous.

6. Presence of an interparietal suture in MNHN LES 17. CT scans reveal a partially patent interparietal suture in NHMUK PV RU B23 that is not apparent in external view. An open suture appears to be present in NHMUK PV R11004.

7. Embayment between the parietals in MNHN LES 17 forms a “sharper V” than in L. diagnosticus (Knoll, 2002a:239), as illustrated by Sereno (1991). Dorsal views of MNHN LES 17 and segmented CT data of NHMUK PV RU B23 do not support this claim.

8. The paroccipital processes of MNHN LES 17 are horizontally-directed, compared to those of NHMUK PV RU B23 as depicted by Sereno (1991). CT scans show that the paraoccipital processes of NHMUK PV RU B23 are more horizontally-directed than illustrated by Sereno (1991); this discrepancy was also noted by Knoll (2002b). Furthermore, the left paroccipital of NHMUK PV RU B23 is slightly dorsally inclined while the right is slightly ventrally inclined, suggesting deformation in this area.

9. The distal ends of the paraoccipital processes are more inflated in MNHN LES 17 than in NHMUK PV RU B23. CT scans reveal an expansion of the distal ends of the processes in the syntype.

10. Basipterygoid processes are less laterally-directed in MNHN LES 17 than figured by Sereno (1991). This does not appear to be the case when comparing segmented basisphenoids from the syntypes and MNHN LES 17 in posterior views.

11. The coronoid eminence is higher in MNHN LES 17 than in L. diagnosticus. Segmented CT scans of the left side of NHMUK PV RU B23 and both sides of NHMUK RU B17 (‘palatal’ block) demonstrate that the coronoid eminence of L. diagnosticus is higher than previously depicted due to the tall coronoid.

12. Shorter retroarticular process of MNHN LES 17 than NHMUK PV RU B23. The quadrate has been anteriorly displaced in the latter; segmentation of CT data from both syntypes demonstrates the retroarticular processes were of similar relative length as in MNHN LES 17.

13. Relatively larger (and more angular) orbit and larger supratemporal fenestrae in MNHN LES 17 than in NHMUK PV RU B23. Both skulls are deformed and the postorbital bars and upper temporal bars in MNHN LES 17 are incomplete, rendering these observations qualitative.

Additional similarities between the L. diagnosticus syntypes and MNHN LES 17 include: a midline groove between the nasals; lack of a sagittal crest in MNHN LES 17, as observed by Knoll (2002a); concave ventral surface of the basisphenoid; and an anteriorly arched caudal margin of the quadrate. Remaining differences between MNHN LES 17 and the syntypes include: the presence of a midline suture between the supraoccipitals in the former (almost certainly due to damage as the supraoccipital is an unpaired element in diapsids); and a gap between the dorsal margin of the supraoccipital and parietal (though such a gap occurs frequently in dinosaurs and may be ontogenetic). Based on the overwhelming similarities between MNHN LES 17 and the syntype skulls, we assign MNHN LES 17 to Lesothosaurus diagnosticus.

A larger partial skull (MNHN LES 18) was also assigned to Lesothosaurus sp. by Knoll (2002b) based on numerous similarities between it and the syntypes. Knoll (2002b) acknowledged that many of the differences between MNHN LES 18 and the syntypes (e.g., relative size and shape of the orbits and fenestrae, shape of the skull roof) could be attributed to dorsoventral crushing of Paris specimen and possibly ontogenetic changes. Other differences identified by Knoll (2002b) include:

1. The right postorbital of MNHN LES 18 is not flush with the frontals as in NHMUK PV RU B23. As noted above, it is likely that all sutural contacts on the external surface of the skull roof were originally flush and such offsets can be attributed to deformation.

2. Direct contact between the head of the quadrate and paroccipital process in MNHN LES 18. Given that the squamosal is missing and that the specimen is strongly crushed, the absence of the postquadratic process of the squamosal between the quadrate and paroccipital process cannot be confirmed and seems unlikely.

3. Lack of a median ridge on the supraoccipital of MNHN LES 18. As noted by Knoll (2002b) the supraoccipital of this specimen is broken at the midline; thus, it is impossible to determine whether or not a nuchal crest was present.

4. Ventrally-directed paraoccipital processes in MNHN LES 18. See comments above as well as noting strong dorsoventral crushing of MNHN LES 18.

The only remaining differences between MNHN LES 18 and the syntype skulls include the presence in the former of a weak sagittal crest and the much larger size of MNHN LES 18 compared to the syntypes. Knoll (2002b) acknowledged that specimens assigned to L. diagnosticus by Sereno (1991) already exhibit a large range of body sizes. Furthermore, recent work has demonstrated that the largest specimen of the early ornithischian Heterodontosaurus is nearly three times the body length of the smallest known individual (Porro et al., 2011). Thus, it is likely that MNHN LES 18 represents a larger individual of Lesothosaurus diagnosticus. Alternatively, MNHN LES 18 could represent the skull of the larger ornithischian Stormbergia (Butler, 2005), also present in the Upper Elliot Formation and presently known from postcranial material only; however, the validity of this taxon is in doubt and there is evidence that it may, in fact, represent an adult Lesothosaurus (Knoll, Padian & de Ricqles, 2009).

Conclusions

Together with visual inspection of specimens, CT-scanning and 3D visualization was used to produce a detailed anatomical description of the skull of Lesothosaurus diagnosticus, revealing new anatomical details such as sutural morphology and internal structures. Elements obscured by matrix or other bones were described for the first time. This new description was used to assign two specimens previously identified as Lesothosaurus sp. MNHN LES 17 (and possibly MNHN LES 18) to Lesothosaurus diagnosticus.

Supplemental Information

Figure S1 3D PDF of Lesothosaurus diagnosticus (NHMUK PV RU B23)

Click here for additional data file.

Figure S2 3D PDF of Lesothosaurus diagnosticus (NHMUK PV RU B17, ‘snout block’)

Click here for additional data file.

Figure S3 3D PDF of Lesothosaurus diagnosticus (NHMUK PV RU B17, ‘palatal block’)

Click here for additional data file.

Figure S4 3D PDF of Lesothosaurus diagnosticus (NHMUK PV RU B17, ‘brain case block’)

Click here for additional data file.

Many thanks to Sandra Chapman (NHMUK) and Daniel Goujet (MNHN) for access to specimens. Hauke Bartsch (Visage Imaging), Jean-Luc Garnier (FEI Visualization Sciences Group), Stephan Lautenschlager (University of Bristol), Justin Lemberg (University of Chicago), Emily Rayfield (University of Bristol), Alejandra Sánchez-Eróstegui (FEI Visualization Sciences Group), Christian Wietholt (Visage Imaging) provided helpful technical advice for Amira/Avizo. Richard Butler (Unviersity of Birmingham) and David Norman (University of Cambridge) provided important insights into early ornithischian anatomy. We thank Ryan Ridgely (Ohio University) for assistance with microCT scanning of NHMUK PV RU B17 and Timothy Ryan, Avrami Grader, and Phillip M. Halleck (Center for Quantitative Imaging, Pennsylvania State University) for assistance with microCT scanning of NHMUK PV RU B23. Clint Boyd (North Dakota Geological Survey) and Peter Galton (University of Bridgeport) provided helpful and constructive reviews of an earlier version of this manucript and we thank the editor Hans-Dieter Sues (Smithsonian Institution) for expeditious handling of the paper.

Institutional Abbreviations

MNHN Muséum National d’Histoire Naturelle, Paris, France

NHMUK The Natural History Museum, London, UK

SAM Iziko South African Museum, Cape Town, South Africa.

Additional Information and Declarations

Competing Interests

Author Contributions

Data Availability

The authors declare there are no competing interests.

Laura B. Porro conceived and designed the experiments, performed the experiments, analyzed the data, contributed reagents/materials/analysis tools, wrote the paper, prepared figures and/or tables, reviewed drafts of the paper.

Lawrence M. Witmer conceived and designed the experiments, analyzed the data, contributed reagents/materials/analysis tools, wrote the paper, reviewed drafts of the paper.

Paul M. Barrett conceived and designed the experiments, analyzed the data, wrote the paper, reviewed drafts of the paper.

The following information was supplied regarding data availability:

Raw data is included as supplemental files.

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
