# Peer review of "Digital preparation and osteology of the skull of Lesothosaurus diagnosticus (Ornithischia: Dinosauria)"

_PeerJ, doi:10.7717/peerj.1494_

## Round 0.1 · original submission · Minor Revisions

Both reviewers provide helpful comments on the manuscript. The addition of scale bars in Figs. 2-11 is mandatory. Reconstructions of the skull in different views are not necessary because the processed scan images show excellent detail of the cranial structure and reconstructions would not add substantial data. I leave it to the authors to decide whether or not they want to include reconstructions. When submitting a revised version of the manuscript please respond in detail to the reviewers' comments.

·

Basic reporting

The only issue that falls under this category is that Figures 2 through 11 need to have scale bars added. Also, are the raw CT data going to be made available anywhere online for other researchers to access? While this certainly isn't required for publication, it would be very beneficial to the community.

Experimental design

No Comments.

Validity of the findings

No Comments.

Additional comments

This is an excellent manuscript that provides a wealth of anatomical information for a taxon that is critical for understanding trends and patterns in early ornithischian evolution. The description is informative and easy to read, and the figures are of excellent quality. I have a few comments on the manuscript, but they are relatively minor. The biggest issue that needs to be addressed is the lack of scale bars in Figures 2 through 11. I realize the authors state in the figure captions that different bones are not to scale, but it is difficult to assess this material without knowing their sizes relative to each other. Without scale bars, these excellent figures will lose some of their usefulness.

Other points to address are as follows:

General Comment: In many places it would be useful to cite the anatomical abbreviation along with the appropriate figure. Doing so would more quickly guide the reader to the appropriate anatomical feature being discussed, without having to look up abbreviations constantly when scanning from the text to the figure.

Line 66: This is a good place to cite Rosenbaum and Padian (2000) because they describe more cranial material for Scutellosaurus. [Rosenbaum, J. N., and K. Padian. 2000. New material of the basal thyreophoran Scutellosaurus lawleri from the Kayenta Formation (Lower Jurassic) of Arizona. PaleoBios 20:13-23.]

Line 78: The citation Maidment et al. (2013) is not in the References section, but there is a citation of Maidment et al. (2014) that is not cited in the text. I assume these are the same citation, and the year just needs to be corrected on one.

Line 93: Is the citation here of Ginsburg (1964) meant to be for the taxonomic authority of Fabrosaurus australis? If so, then taxonomic authorities need to be cited for all other taxa mentioned in the text at their first appearance. I think the only other taxon that has the taxonomic authority presented with it is Lesothosaurus diagnosticus.

Lines 198-201: I am not sure what evidence is being used to support the observation that the external narial opening “…potentially extending posteriorly to overlie the maxilla for a short distance.” What specimen is this based on? I don’t think it is one figured in this paper. This observation needs at least a specimen citation, and preferably a citation for a publication and figure that shows this condition if its not one of the specimens figured in this paper.

Lines 241-244: The articulation surface for the maxilla discussed in this section is a bit confusing. Looking at Figure 3 (a.mx), the location where this articulation facet is labeled seems to be the base for the missing maxillary process. Is the label in the wrong place? If not, where would the maxillary process attach? Labeling both in Figure 3 would help to clarify the placement of that articulation facet.

Lines 283-289: Included in this list of structures found in this same area of the maxilla in other taxa should be the maxillary fenestra of Hypsilophodon and Haya.

Line 296: On the maxilla, I think the contact for the jugal should be dorsomedially directly to match the ventrolaterally directed articulation surface on the jugal. In the current version, the contact on both the maxilla and the jugal are both described as ventrolaterally directed.

Line 298: Here you called this the “posterodorsal process of the premaxilla,” but earlier in the manuscript you call it the “maxillary process.” Change it here for consistency.

Lines 533-536: Can you label the fenestrae/openings discussed here on Figure 7?

Lines 623-624: Amongst basal ornithischians, basal neornithischians, and basal ornithopods, what taxa are known to fuse the braincase elements other than the exoccipital/opisthotic, basisphenoid/parasphenoid, and the paired parietals? Either all known specimens of these taxa are juveniles, or fusion of the remainder of the braincase is extremely delayed. I’m not sure fusion of the braincase is the best feature to use in these taxa to assess sexual or somatic maturity. However, if I’m missing something here and it does occur in some taxa, please cite that information here to support this claim.

Line 664: The figure citation should be 9D, not 9B.

Line 817: A space is needed between the semicolon and the Thulborn (1970) citation.

Line 821: I think the contact surface for the splenial should be described as on the ventromedial surface of the dentary, not the ventral surface, or does the splenial wrap underneath the dentary?

Line 884: The statement “The coronoid it is preserved…” should be “The coronoid is preserved…”

Lines 849-851: This sentence describes the prearticular contacting the prearticular. I think the first reference to the prearticular is supposed to be the coronoid.

Line 861: I think the figure citation should be 11F, not 11G.

Line 877: The word “anteriorly” should be “anterior”

Lines 887-888: The wording “internal surfaces” is a bit confusing. I am not sure if this just means medial in general, or the surfaces that face into the Meckelian canal.

Line 906: Should this be the anterodorsal margin of the internal mandibular fenestra, or am I visualizing this incorrectly?

Lines 1128-1129: The Irmis et al. (2007) citation is not in the text of the manuscript, so is not needed in this section.

Figure 2 and caption: In this figure and the caption you label the exoccipital (ex), but in the text you use the term otoccipital. I would stay with the term otoccipital for the sake of consistency. Also, the abbreviation EMF is used in the figure but not defined in the caption. Finally, in the figure some of the labels are in bold text, while others are not.

Figure 3: See above comment on the location of the articulation surface for the maxilla.

Figure 4 and caption: In the caption, the right maxilla in dorsal view should be figure part E, not D.

Figure 5 caption: I think figure parts M and N are a left jugal, not a right, unless the images are reversed. Additionally, figure parts O, P, and Q are a left postorbital, not a right. The abbreviation OB should be OT, and it is out of alphabetical order. The abbreviation RUG is not defined. The exoccipital is referenced again and it should be otoccipital.

Figure 6 caption: The left quadrate should be figure parts D, E, F, and G, not C, D, E, and F. The second sentence about the prefrontal is not relevant here and should be removed. In the third sentence, it should be the left palpebral, not the eft prefrontal. Again, the exoccipital is referenced when it should be otoccipital.

Figure 7: Use of more colors here would really help differentiate the bones.

Figure 11 caption: The abbreviations CMJ and EMF are not defined.

If you have any questions regarding these comments, please feel free to contact me.

Clint Boyd

·

Basic reporting

No comments

Experimental design

No comments

Validity of the findings

No comments

Additional comments

This is an excellent well written MS and I've provided minor editorial-type comments on the doc copy of the MS (converted to a pdf file, with suggestions marked in red).

Various corrections are provided regarding the description of the skull of Lesothosaurus given by Sereno (1991) but, despite the details given in this MS, Sereno (1991, figs 11, 12, 13) still provides the only overall reconstructions of the skull in lateral, dorsal, palatal and occipital views, of the braincase in lateral view, and of the mandible in lateral, medial and dorsal views. Updated versions of these views that incorporate the corrections noted in the text should be added to this MS.

To be fair to the authors, this MS follows what appears to be a trend in many recent papers on the cranial anatomy of basal ornithischian and sauropodomorph dinosaurs, the tendancy to figure the specimens as preserved with no reconstructions in lateral view (or in any other one).

---

## Round 0.2 · accepted · Accept

The authors have taken care to respond to all of the referees' suggestions. I am pleased to recommend the revised manuscript for publication.